



# The Meteorology and Chemistry of High Nitrogen Oxide Concentrations in the Stable Boundary Layer at the South Pole

William Neff [1], Jim Crawford [2], Marty Buhr [3], John Nicovich [4], Gao Chen [2], Douglas Davis [4]

[1] NOAA/ESRL Physical Sciences Division and University of Colorado Cooperative Institute for
Research in Environmental Sciences, Boulder CO 80305, USA

[2] NASA Langley Research Center, Hampton, VA 23681, USA

[3] Air Quality Design, Golden, CO 80403, USA

[4] School of Earth and Atmospheric Science, Georgia Institute of Technology, Atlanta, GA 30332, USA
(D. Davis submitted posthumously)

*Correspondence to:* William Neff (william.neff@noaa.gov)

**Abstract.** Four summer seasons of nitrogen oxide (NO) concentrations were obtained at the South Pole during the Sulfur Chemistry in the Antarctic Troposphere (ISCAT) program (1998 and 2000) and the Antarctic Tropospheric Chemistry Investigation (ANTCI) in (2003, 2006-7). Together, analyses of their data here provide insight into the large-to-small scale meteorology that sets the stage for high NO and the significant variability that occurs day-to-day, within seasons and year-to-
year. In addition, these observations reveal the interplay between physical and chemical processes at work in the stable boundary layer of the high Antarctic plateau. We found a systematic evolution of the large-scale wind system over the ice sheet from winter to summer that controls the surface boundary layer and its effect on NO: Initially in early spring (Days 280-310) the transport of warm air and clouds over West Antarctica dominates the environment over the South Pole; In late spring (Days 310-340), of significance to NO, the winds at 300-hPa exhibit a bimodal behavior alternating between NW and
SE; In early summer (Days 340-375), the flow aloft is dominated by winds from the Weddell Sea. During late spring, winds aloft from the SE are strongly associated with clear skies, shallow stable boundary layers, and light surface winds from the east: it is under these conditions that the highest NO occurs. Examination of the winds at 300 hPa from 1961 to 2013 shows that this seasonal pattern has not changed significantly although the last twenty years have seen an increasing trend in easterly surface winds at the South Pole. What has also changed is the persistence of the ozone hole, often into early
summer. With lower total ozone column density and higher sun elevation, the highest actinic flux responsible for the photolysis of snow nitrate now occurs in late spring under the shallow boundary layer conditions optimum for high accumulation of NO. This may occur via the non-linear $HO_X$-$NO_X$ chemistry proposed after the first ISCAT field programs and $NO_X$ recycling to the surface where quantum yields may be large under the low-snow-accumulation regime of the Antarctic plateau. During the 2003 field program a sodar made direct measurements of the stable boundary layer depth
(BLD), a key factor in explaining the chemistry of the high NO concentrations. Because direct measurements were not available in the other years, we developed an estimator for BLD using direct observations obtained in 2003 and step-wise linear regression with meteorological data from a 22-m tower (that was tested against independent data obtained in 1993). These data were then used with assumptions about the column abundance of NO to estimate surface fluxes of $NO_X$. These results agreed in magnitude with results at Concordia Station and confirmed significant daily, intraseasonal and interannual
variability in NO and its flux from the snow surface. Finally, we found that synoptic to mesoscale eddies governed the boundary layer circulation and accumulation pathways for NO at the South Pole rather than katabatic forcing. It was the small scale features of the circulation including the transition from cloudy to clear conditions that set the stage for short-term extremes in NO whereas larger-scale features were associated with more moderate concentrations.



## 1.     Introduction

The Investigation of Sulfur Chemistry in the Antarctic Troposphere (ISCAT) field programs in the austral summers of 1998 and again in 2000 (Davis et al. 2004a; Davis et al. 2001) discovered unexpectedly high atmospheric nitrogen oxide (NO) concentrations at the South Pole (SP).   These early investigations suggested that the high NO levels were associated with continuous sunlight, shallow stable boundary layers, downslope flow from the east Antarctic plateau, and associated non-linear $NO_X$-$HO_X$ chemistry that resulted in long $NO_X$ lifetimes, (Davis et al. 2004a; Davis et al. 2008)).  It was also argued that the high levels of NO at the SP, compared to other polar sites, were partly due to the long fetches for air parcels in the katabatic flow from the Antarctic high plateau and the lack of a diurnal cycle, thus allowing continuous photolysis of snow nitrate.  A major field program followed in 2003 to re-examine this phenomenon, the Antarctic Tropospheric Chemistry Investigation (ANTCI).   This new study involved ground-based measurements at SP as well as aircraft probing over more extensive areas of the Antarctic Plateau in 2003 (Davis et al. 2008) and then again in 2005 (Slusher et al. 2010).  These field studies reinforced the earlier ISCAT results and introduced evidence suggesting that fast recycling mechanisms for redeposited nitrate could be important given that photolysis of buried nitrate appeared inadequate to explain the sustained levels of atmospheric $NO_X$. In combination with meteorological factors, this recycling enabled $NO_X$ levels which routinely reached and/or exceeded several hundred pptv.

Because of significant daily, seasonal, and inter-annual variability in NO observed in the previous field studies this paper will examine in greater detail the influence of large- to small-scale meteorological circulations on the boundary layer and associated chemical factors that set the stage for high NO episodes. A key feature of the boundary layer at the South Pole is that it is statically stable throughout most of the year and often very shallow which leads to the confinement of surface chemical emissions near the surface except during higher-wind events and cloudy periods.  Important factors also influencing the near-surface chemical environment include the winter-to-summer seasonal cycle as well as inter-annual variability  and synoptic variability in cloudiness (important to the surface energy budget and the boundary layer structure), low snow accumulation, winds, temperature, total column ozone and the radiative changes associated with the breakup of the stratospheric polar vortex and ozone hole in the spring.  This effort uses previous data sets complemented by a new unreported data set collected in 2006-07. The foci here are the origin of high NO episodes and their relationship to stable boundary layer physics and large scale dynamics as well as the unique chemical processes at work on the high plateau.

## 2.     Boundary layer meteorology, NO and the seasonal cycle

First we examine the meteorological factors controlling boundary layer depth (BLD) and NO variability, over hourly, seasonal and interannual time scales, using the summer field program data from 1998, 2000, 2003, and 2006-07. Because NO was the one species measured consistently in all field studies cited, it will be used in most of the follow-on discussions (NO and $NO_2$ were both measured in 2006-2007).  This analysis benefits from a repeatable weather cycle at SP and the lack of a diurnal solar insolation cycle. The weather cycle typically includes a period of advection of warm air, higher wind speeds, and clouds with a deep boundary layer followed by clearing skies, strong radiative surface cooling and formation of shallow statically stable boundary layers. A unique aspect of the SP includes a high snow albedo and  low sun elevation angle. Because of this combination the surface typically suffers a net loss of energy under clear-sky conditions (Stone and Kahl 1991) even at the summer solstice.  This effect is in contrast with summer observations at Concordia Station (~75°S, 3233 m



ASL) and Dome Argus (~80°S, 4093 m ASL) that show a diurnal cycle of thermal convection followed by the formation of a stable boundary layer during the "evening (low sun elevation)" period (Frey et al. 2015) (Bonner 2015; Bonner et al. 2010).

The synoptic weather variability at SP occurs in the context of a strong seasonal cycle from early spring to summer as the sun rises and the omnipresent surface inversion weakens to less than 10% of its winter value (Neff 1999) implying only weak katabatic influence during the Austral summer (discussed in more detail below). Also of interest are the major shifts in
large-scale weather patterns that occur from early-October to mid- or late-December, coincident with the weakening of the circumpolar trough, warming of the stratosphere and breakup of the polar vortex and the stratospheric ozone hole. The dynamical consequences of this transition also affect boundary layer behavior that is dependent on cloud fraction, surface radiative balance, inversion strength, as well as upper-level winds and associated surface pressure gradients. In our discussions we will use the grid direction convention at SP where grid north is aligned with the Greenwich Meridian (Fig. 1).

**2.1      The large-scale meteorological environment**

Boundary layer and inversion depths as well as surface wind speed and direction at SP have been shown to be controlled to a large extent by local-to-large-scale terrain and large-scale weather patterns (Neff 1980, 1986; Neff 1999). At 14.0 x $10^6$ km$^2$ and somewhat pole centric, the major storm tracks lie north of the Antarctic continent. This is in contrast to the Greenland ice sheet with an area of about 1.8 x $10^6$ km$^2$ which lies close to and often within the northern storm track. The SP is at a
comparable elevation (~2.8 km) to Summit Station Greenland (~3.2km).

**2.1.1      The seasonal cycle, upper-level winds and their influence on surface flows**

As the stratosphere over Antarctica begins warming around the time of the spring equinox, prevailing winds at the height of the summer tropopause over the South Pole (~300 hPa) evolve in three stages as shown in Fig. 1 (based on rawinsonde data for the period 1990 to 2010). In Early Spring (JD280-310), 300-hPa winds are largely from the direction of west Antarctica
and associated with high cloud fraction and advection of warmer air at the South Pole (Neff 1999). This period coincides with the maximum in the semi-annual oscillation when the circumpolar trough is most intense (Meehl 1991). The increase in synoptic weather activity at this time suggests increased likelihood of transport from the ocean, surrounding sea ice and more northerly latitudes to the interior of the continent. For austral winters Neff (1999) calculated isobaric geostrophic temperature advection at the SP from averaged rawinsonde profiles and found advective warming associated with NW to SW winds aloft
and cooling associated with SE winds aloft. As the season evolves during the Late Spring (JD310-340), the winds become bimodal as shown in Fig. 1 where percentages in 45°-bins have peaks centered 180° apart at 157.5° and 337.5°. The most significant aspect of this bimodal distribution is the much diminished transport of warm air from west Antarctica and lower cloud fraction at SP. During the Late Spring period 43% of the 300 hPa winds lie between 90° and 225° whereas 31% lie between 270° and 360°. As reported in Neff (1999), this Late Spring period also has the lowest average cloudiness (see also
Fig. S8 for more recent data). Furthermore, for winds greater than 10 ms$^{-1}$ at 300 hPa, between 1990 and 2010, in the 45° intervals centered on 157.5° and 337.5°, the average surface inversion strength (from rawinsonde soundings) and surface wind speed are 6.3°C and 3.5 ms$^{-1}$ and 2.5°C and 5.4 ms$^{-1}$, respectively. Thus, in addition to reduced cloudiness during this period, surface wind speeds are less and surface-based inversion strengths are larger for 300-hPa winds from the SE. (For the Early Summer period, the results are similar at 2.2°C and 3.1 ms$^{-1}$ and 0.3°C and 5.9 ms$^{-1}$, respectively). With the higher
frequency of winds from the northwest in early summer with increased cloudiness, low static stability and higher surface



wind speeds, high NO should be less likely on the average, after mid-December. Conversely, periods of low-cloud fraction are consistent with increased radiative cooling of the surface and the stronger surface temperature inversions that result are consistent with higher NO.

The difference in surface wind speed and inversion strength can be explained in terms of the 300 hPa geopotential height

(GPH) patterns corresponding to the NW-SE modes of upper level winds (Fig. S1). For NW winds, low GPH lies between the Ross and Weddell Seas over West Antarctica with higher GPH over the continental interior. For SE winds, lower GPH lies in the eastern hemisphere extending over most of the high Plateau. Although somewhat different than the winter pattern in Neff (1999) where for SE winds aloft a strong ridge of high pressure extended from the Ross Sea over portions of east Antarctica, the large scale pressure gradient force still reveals either an on-shore or off-shore orientation in the two cases. In

the case with low pressure centered mostly over high terrain, downslope flows would be less likely whereas the inverse would be true for off-shore pressure gradient forces.

The frequency of occurrence of SE winds at 300 hPa can, however, vary from year to year and from decade to decade as reported in (Neff 1999). Figure 2 shows the year-to-year variability in the occurrence of SE winds for the more recent period 1990-2013 which except for outliers in 1999 and 2002 (the year of the only sudden stratospheric warming observed over

Antarctica) follows the almost-decadal variability discovered earlier. For each of the four observational periods (approximately mid-November through December for consistency) the frequencies of occurrence of winds at 300 hPa from the SE were as follows: 27% in 1998, 2% in 2000, 24% in 2003, and 42% in 2006. For each of these periods when NO was recorded at SP, the number of hours of NO>250 pptv were as follows: 42% in 1998, 4% in 2000, 56% in 2003, and 29% in 2006. The lack of a one-to-one relationship between the occurrence of SE winds and NO concentrations suggest additional

year-to-year variability in chemical source terms and the details of the synoptic weather conditions favorable to high NO. What does stand out in 2000, with very low frequency of winds from the SE quadrant is the limited number of hours when NO exceeded 250 pptv, thus indicating the seasonal dependence of NO concentrations on the large scale weather patterns over the Antarctic.

Winds aloft from the SE quadrant often coincide with a transition to easterly winds at the surface. Figure 3 shows that these

easterly surface winds were most prevalent in 2003 and 2006, intermediate in 1998, and minimal in 2000. Also, averaged over the four field seasons, highest NO occurred when surface winds were from the east to southeast in concert with winds aloft from the SE. Figure S2 shows the distribution of winds at 300 hPa when surface winds are greater than 2 ms$^{-1}$ and lie between 70$^o$ and 210$^o$ during November and December, from 1961 to 2010. 67% of the winds at 300 hPa are between 90$^o$ and 180$^o$ for these surface wind directions implicating SE winds at tropopause level as a major determinant for easterly winds

at the surface and extreme NO concentrations. Figure S3 explores the relationship of winds aloft to surface wind behavior for average surface wind directions (60$^o$ to 180$^o$) associated with average NO greater than 200 pptv for Late Spring (Days 310-340) and for the full period of rawinsonde data starting in 1961. Figure S3a reveals strong interannual variability in the occurrence of surface winds with an easterly component as well as an upward trend exceeding 10% beginning in the late 1990s. In this case, we are only looking at the period of Late Spring whereas in Fig. 2, the data extended to the end of

December. Figure S3b shows similar interannual variability in SE winds at 300-hPa as well as multi-decadal variability but no systematic trend. In Fig. S3c, we show that extremes in the frequency of occurrence of winds from the SE at 300 hPa





correspond to extremes in the occurrence of surface winds from the SE with $r^2$=0.4. Comparing Fig. S3 with Fig.2 which spanned the period mid-November to the end of December, shows that easterly surface winds occurred primarily after mid-December in 1998: However, in that year, the ozone hole breakup was delayed until the end of December and hence optimum
boundary layer conditions favorable for high NO occurred with still-high actinic flux.

The surface inversion strength also varies strongly with the season as shown in Fig. S4a where we have plotted the average inversion strength from 1961 to 1998 (Neff 1999) as well as daily data for 2006 showing typical short-term variability. (The inversion strength was determined from the daily or twice-daily rawinsonde temperature profile, measuring the difference from the 2-m tower temperature to that at the first relative maximum in the rawinsonde profile). Because of the common
assumption that the winds over Antarctic are largely katabatic in nature, motivated mainly by wintertime observations in coastal areas, we also show in Fig. S4a the geostrophic wind equivalent of inversion strengths of $4^{o}$C (1 ms$^{-1}$) and $18^{o}$C (5 ms$^{-1}$) assuming a terrain slope of 0.001, characteristic of the interior of the ice sheet and derived from (Ball 1960). From the figure, geostrophic winds aloft of only 1 ms$^{-1}$ will dominate over katabatic forcing from mid-December to mid-January. In Fig. S4b we show the distribution of wind speeds at 300 hPa for December, averaged from 1998 to 2014. This figure shows
that almost all synoptic scale winds in December at 300 hPa imply surface synoptic pressure gradients substantially greater than the katabatic ones associated with summer inversion strengths. The conclusion from the previous figures is that synoptic and mesoscale pressure gradients at the surface dominate over those driving katabatic flows over the interior of Antarctica, especially in the period November-January and that 300-hPa winds from the SE provide a significant control over the boundary layer depth important to high NO. Past work has suggested that katabatic trajectories may have been responsible
for long fetches over which NO could accumulate (Davis et al. 2004a). However, from these arguments it appears more likely that meso- to synoptic-scale circulations define accumulation pathways over the interior of Antarctica in the late spring and summer and that these pathways vary significantly on intraseasonal to interannual time scales. Furthermore, as can be seen comparing surface wind directions for maximum observed NO in Fig. 3 with the large-scale topography in Fig. 1b, the surface flows align largely along constant topographic contours between South Pole and Concordia Station (along $123^{o}$E)
rather than downslope. Of note, high NO, up to 400 pptv, along $110^{o}$ E was found for several hundred km from the SP on 4 December 2003 in aircraft measurements (Davis et al. 2008). However, this was an unusual period with a deep boundary layer, high winds as well as high NO.

### 2.1.2    The spring breakup of the polar stratospheric vortex

The warming of the stratosphere between October and December is important for changes in the large scale circulation as
well as to boundary layer behavior at the South Pole (Neff 1999). It has been noted by others that the final warming of the southern stratosphere has a significant influence on the tropospheric circumpolar circulation (Black and McDaniel 2007) and over the interior of the continent (Neff 1999). Furthermore, with the final warming, downward coupling (at Wave-One scales) from the stratosphere to the troposphere ceases (Harnik et al. 2011) although the subsequent impact on the circumpolar storm track has not been explored. The timing of this final warming varies from year to year and is marked
initially as the time the zonal wind in the polar vortex between 30–10 hPa and $54^{o}$S–$75^{o}$S first goes to zero. From the work of Harnik et al. (2011) this occurred very late in 1998 and very early in 2000; 2003 and 2006 followed the long-term trend





toward a later warming. Of primary importance in the delay of the vortex breakup is the persistent low total column ozone and resulting increase in actinic flux later into spring (as in 1998) responsible for increased photolysis of snow nitrate.

In Fig. 1 we looked at the period 1990-2010, the decades following the onset of rapid stratospheric ozone depletion beginning in the 1980s. This depletion resulted in a 30-day delay in the formation of the thermal tropopause over the South Pole as documented in Fig. S5. Prior to 1980, the tropopause formed, on the average, around Julian Day 306 (November 2). After 1990, the average time is Julian Day 336 (December 2), a delay of 30 days. Past analyses calculated the covariance of fluctuations in the easterly component of the surface wind and temperature (essentially the easterly flux of cold air: v'T') from 1961 to 1997 (Neff 1999) finding a persistence of about 20 days later between 1961-1970 and 1988-1997 commensurate with the delay in the breakup of the polar vortex. This suggests that a delay in the breakup of the vortex leads to an increase in easterly surface flows associated with higher NO later in the spring.

Figure S6 compares the seasonal change in total column ozone between 1964-1980 and 1990-2010 (measured with a Dobson Ozone spectrophotometer – available from www.esrl.noaa.gov/gmd/ozwv/dobson/) showing reductions of about 30% in early December. A consequence of delays in recent decades, the actinic flux has increased in the spring as a result of a persisting ozone depleted column coupled with a higher sun elevation angle. Figure 1 suggested that during the late spring/early summer period of higher sun elevation angle and thus higher potential actinic flux, the late-spring period had an almost 50% higher occurrence of SE winds compared to early summer. However, early summer was dominated by NW winds which are typically associated with higher wind speeds, clouds, deeper boundary layers and weaker inversions thus negating the effect of a higher sun elevation in general.

A question remains, however, is whether the distribution of winds aloft in Fig. 1 is an artifact of coincidental changes in the timing of the vortex breakup and the normal seasonal cycle in the circumpolar circulation. Figure S7 explores this question in more detail where we examined winds at both 200 hPa and 300 hPa. In Fig. S7 a,b for Days 270-305 (~October), the wind distributions are very similar to those of Fig. 1 for both 1960-1980 and as well 1990-2010. In Fig. 7S c,d for Days 305-340 (~November) the bimodal distribution seen in Fig. 1 appear at both 200 and 300 hPa as well as in both time periods although the trend is toward a more bimodal distribution at 200 hPa. This suggests that the wind aloft becoming bimodal in November is an artifact of the seasonal cycle rather than resulting from the breakup of the polar vortex. What is different in the most recent decades is the increase in actinic flux at the same time that the seasonal cycle in boundary layer meteorology is more conducive to high NO levels. In Fig. S7 e,f for Days 341-365 (~December), winds at 300 hPa from the northwest are favored as in Fig. 1 with the effect a bit more evident in 1990-2010 compared to 1960-1980. However, at 200 hPa the wind direction distributions are significantly different with a reversal of the distributions between the two 20-year periods. This may be due to the stronger tropopause strength in the 1960s and 1970s (average difference of 6.7°C between 300 hPa and 200 hPa) versus recent decades (1.6°C) that allows winds in the lowermost stratosphere to be less isolated from those in the upper troposphere.

### 2.1.3 Seasonal variation in cloudiness

Shallow boundary layers occur most often with clear skies. With cloudy conditions the boundary layer is usually deep and well mixed (Neff et al. 2008). It has been noted that cloudiness has generally increased over Antarctic in recent decades (Neff 1999; Schnell et al. 1991). Figure S8 shows an updated version of Plate 5 in Neff (1999) from 1957 to 2013 of average



cloud fraction at the South Pole from winter to summer. The figure shows a continued increase in cloudiness, a somewhat decadal periodicity, a mid-summer minimum generally during Julian Days 325-350 as well as a late winter increase in cloud

fraction in the most recent decade. The minimum in cloudiness during Julian Days 325-350, when shallow boundary development is more likely due to increased clear-sky radiative cooling, also coincides with increased actinic flux due to the persistent ozone hole of recent decades.

However, the average cloud fraction can vary within a season and from year to year. For example, the average cloudiness for days 310-340 (late spring) was 6.5, 5.1, 3.9, 5.1 for 1998, 2000, 2003, and 2006 respectively. The relative minimum in

cloudiness marks the transition in the dynamical regime favoring transport of clouds from West Antarctica to one favoring transport from the area of the Weddell Sea. Following this minimum, cloudiness again increases in concert with decreasing inversion strength. This period also occurs at the time of the sea ice minimum implying a shorter moisture transport path from the open ocean to the interior. Because the total column ozone increases at the same time, the lower actinic flux coupled with higher cloud fraction and its effect on the boundary layer will not be as favorable for high NO concentrations.

Figure 4 shows an example of this behavior for 2006 when a minimum in cloudiness occurred around JD 337 (3 December 2006) just before the photolysis rate for nitrate, $j(NO_3^-)$ started decreasing by 40%. This decrease was associated with the breakdown of the winter stratospheric vortex and the attendant increase in ozone which results in decreased UV flux due to absorption in the stratosphere. Figure 4a summarizes the consistent timing of this cloud minimum showing the 2000-2010 average observer-based cloud-fraction compared to that in 2006 (using 14-day running averages). Figure 4b, using three-day

running averages, shows the timing of the cloud-fraction minimum in 2006 relative to the start of rapid warming and ozone increases in the lowermost stratosphere: the maximum in nitrate photolysis rate occurs just prior to the stratospheric warming as shown in the figure as well as in Fig. 4c. Figure 4c shows the direct solar irradiance for JD 335-340 (December 1-5) where decreases indicate the presence of clouds which affect the surface radiation balance. The daily average cloud fraction (green bars - from the SP climatological record) shows close inverse agreement with the average direct solar irradiance. Starting on

JD 337, skies clear, leading to the rapid formation of a surface temperature inversion below 10m as seen on the 22-m tower. Several hours after the cooling starts, the winds shift from N to SE (winds not shown), the inversion strengthens and hourly NO increases to nearly 800 pptv (1200 pptv in 10-min averages) despite a modest decrease in actinic flux. What is interesting in this case is the fact that even though the photolysis rate is decreasing, NO is increasing locally. What is unknown is whether high NO was generated earlier away from the SP and then transported into the site. This will be discussed more

fully in Section 3.2. These results still suggest the dominant role of a strong surface inversion, shallow boundary layer and potential non-linear $NO_X$ chemistry as suggested by Davis et al. (2004).

## 2.2    The role of local topography

Figure 5 shows the local terrain within several hundred km of the South Pole as well as the location of the clean air sector ($340°$ to $110°$ - where no potential contamination sources are allowed) and other areas of potential anthropogenic $NO_X$

sources ($210°$ to $340°$) that depend on wind direction. As noted in the figure, the station power plant lies almost due west of the Atmospheric Research Observatory (ARO) whereas runways and aircraft taxiways and parking areas encompass much of the $210°$ to $340°$ sector. To avoid potential contamination, data from this sector were removed from the analysis. Except for



the power plant, other anthropogenic $NO_X$ sources are likely to be intermittent and show up as large outliers and were also removed.

Although discussions of the slope flows at the SP generally assume an idealized sloped plane of great fetch, the local topography is much more complex as seen in Fig. 5. In fact, the terrain rises 250 m in the first 150 km to the east of the SP (slope~0.002) just adjacent to an extensive plateau (slope~0.0003) that begins 150 km east of the SP and extends 200 km to the foot of the relative steep central Antarctic dome that rises to 4 km ASL to the northeast. In addition, a small "air drainage" basin about 100 km in extent lies to the southeast. The average surface wind direction from the northeast follows

terrain isopleths for several hundred km rather than flowing down a slope as is normally envisioned. However, while the average surface wind direction at SP is from the northeast along the terrain, the wind direction actually fluctuates between north (with warmer temperatures) and east (with colder temperatures). In the winter, the difference in the centers of the wind direction distributions for warming- and cooling-events is $90^o$ (Neff 1999). The distribution for our summer experimental periods is narrower with maxima at $20^o$ and $70^o$ and a relative minimum at $40^o$. Figure S9 shows an example of the

distribution of 10-min NO concentrations as a function of wind speed and azimuth for November 16 -30, 2006. In this case surface wind directions between $50^o$ and $150^o$ and speeds less than 5 ms$^{-1}$ account for most of the high NO concentrations, consistent with Fig.3. These directions align closely with the shallow basin to the east-northeast and southeast of the South Pole (Fig. 5).

### 2.3    Possible meteorological precursors to high NO

We have shown the dependence of NO on 10-m wind speed and direction (Figs. 3 and S9). However, in both 2003 and 2006, prolonged and high NO occurred in the last half of November (Fig. S10). This raises the question of whether antecedent conditions set the stage for high NO. In particular, past studies (Mahesh et al. 2003) have noted the systematic change in wind speed between November and December and also concluded that blowing snow was prevalent at the South Pole when the wind exceeded about 7.5 ms$^{-1}$ which was more likely in November. Figure S11 shows the 1-min wind speed distributions

between November and December for the four ANTCI seasons revealing the much higher prevalence of winds in excess of 7.5 ms$^{-1}$ in November. Table S1 shows a more detailed breakdown of the number of hours of winds exceeding 7.5 ms$^{-1}$ for October, November, and December for each of 1998, 2000, 2003, and 2006. Unlike the other field experiment years, 2003 saw the fewest high winds in November and the greatest number in December (although light winds only prevailed after 10 November). Snow stake-field depth data in 2003 showed an increase in snow depth of 6.1 cm in October whereas November

showed a decrease of 1 cm (from Local Climatological Data records - ftp://amrc.ssec.wisc.edu/pub/southpole/climatology/). A coincidence between blowing snow and increased surface nitrate has been observed at Halley Station (Mulvaney et al. 1998) where high wind events in November coincided with increased accumulation followed by a lowering of the snow surface though processes such as sublimation and metamorphosis such as also observed in 2003 at the SP. Following the high wind period at Halley Station, a factor-of-four increase in surface snow nitrate concentrations was seen (followed by

significant losses over the subsequent 10 days). They suggested one possibility for this increase was scavenging by wind-blown snow crystals of air rich in nitric acid during the high wind period followed by the loss of water vapor via sublimation resulting in higher concentrations of nitrate on the surface. Recent research at Concordia Station on the high Antarctic Plateau reported a similar increase in surface nitrate following a period of high winds at the end of November 2011 (Berhanu





and Co-authors 2015) followed by decreases coincident with strong fluxes of nitric oxide from the snow to the atmosphere
during an extended period of lower wind speeds (Frey et al. 2015). Whether such sequences are coincidental or have an
underlying physical/chemical explanation is yet to be determined.

### 3. Case studies and insights from new 2006 data

Here we focus on events that reflect the coupling of boundary layer meteorology and chemistry using the unpublished data
set obtained from 2006 to early 2007. Of the four field periods, 2006-2007 was the only one to capture the early spring to
summer transition in meteorology and chemistry. To provide an overview of the seasonal evolution and short-term weather
events on near-surface NO concentrations in 2006, Fig. 6 shows the time series of NO, wind direction, wind speed, and
temperature from 1 November 2006 (Day 305) through 13 January 2007 (Day 378) with eight events highlighted and labeled
"A" through "H". In addition, hours of relatively clear skies (direct irradiance >80% of maximum) are indicated on the
temperature plot with hash marks plotted at -20 C and daily cloud fraction less than 6/10$^{ths}$ at -15 C on the temperature time
series panel. In general, high direct irradiance associated with a lack of clouds, implies a net loss of radiation from the surface
because of the cold dry atmosphere that prevails over the SP, high albedo, and the lower sun angle compared to other polar
sites.
In each of these cases in 2006, higher NO follows the clock-wise rotation of the wind past $60^{o}$ together with slight-to-
moderate drops in temperature (see Fig. 4). The duration in each case ranges from less than a day to multiday periods.
Episodes "A to C" follow high wind episodes by 2 to 3 days. Episode A in Fig. 6 reveals a rapid increase in NO as the wind
shifts to the east and weakens, followed by a rapid decrease as the wind increases to 10 ms$^{-1}$ and shifts to northerly. Episode
"B" follows several periods of winds reaching 10 ms$^{-1}$ until JD320 (16 November) followed by lighter winds from easterly
directions and persistently high NO for almost ten days. During the period of high NO in "B", winds at 300 hPa were
consistently from the SE, an artifact of a persistent, elongated trough of low pressure over the higher terrain grid NE of the
South Pole (see also Fig. S1). Episode "C" was explored in the example shown in Fig. 4 and will be discussed more fully
below. Episodes "D-G' represent short excursions of NO and all occur with lower wind speeds and occasional easterly
directions. Episode "H" is included because of its contrast with early periods and its occurrence as one of the few "events"
available during the summer period: "H" begins with the temperature exceeding -20 C and then dropping to between -30 C
and -25 C with the arrival of a cold front and clear skies. The highest temperatures, above -20$^{o}$C, occurred during totally
overcast skies, a weak inversion, and NO concentrations less than 100 pptv. Later, there were periodic wind shifts to the
east, clear skies, light winds and colder temperatures, and higher but modest levels of NO (200-300 pptv). In this case, NO
levels remained only moderate despite weak easterly winds, increased inversion strength, a shallow boundary layer, and a
longer potential fetch.

### 3.1 NO levels before and after the final warming of the stratosphere

Of particular interest in 2006 is the fact that Episode 'C" occurred just prior to the final warming of stratosphere (Fig. 4). As
noted earlier, from 1990 to 2010, this warming occurs on the average on Day 336 and coincides with the transition in the



seasonal cycle as summarized in Fig. 1. Comparing NO levels for 20 days before and after Episode "C", 20% of hourly NO
levels >400 pptv occur before (JD305-340) and only 6% occur after (JD340-360). Similarly, in 2003, 43% of hourly NO
levels >400 pptv occur before (JD326-335) and only 10% occur after (JD340-360). In both 2006 and 2003 high NO events
were typically associated with periods of SE winds aloft. Table S2 shows the breakdown of SE (45°-225°) and NW (225°-
360°) winds at 300 hPa. For both 2003 and 2006 the distributions of wind directions for JD<340 are very similar but the
average NO differs by a factor of two (2003: 445 pptv; 2006: 247pptv). For JD>340, 2003 has many more NW winds than
2006 but similar average NO (200 pptv in 2003 and 175 pptv in 2006). In 2003, 300-hPa wind distributions were consistent
with long-term averages (1990-2014) whereas 2006 was an anomaly in early summer with many fewer NW winds aloft and
many more from the SE (see Fig. 2). An implication from the 2006 data is that despite meteorological conditions (wind
speed and direction) conducive to high NO in both November and December, emission fluxes from the snow are likely lower
in December, consistent with the results from (Frey et al. 2015) for the last half of December and early January at Concordia
Station. Frey et al. argues that the snow nitrate is composed of both photo-stable and photo-labile fraction and that the photo-
labile fraction decreased later in December at Concordia Station. Our results would be consistent with that interpretation
although one would also need to consider the effect of warmer and cloudier conditions with deeper boundary layers later in
the summer as the seasonal cycle progresses.

It should be noted that we did not consider 1998 or 2000: in 1998, the breakup of the vortex was much delayed whereas in
2000 it was very early (Harnik et al. 2011; Neff 1999). Figure S10 summarized the spring cycle in NO levels and extremes
showing hourly NO concentrations for all four years as well as the day on which total column ozone at the South Pole first
exceeds 300 DU. The figure showed that 2003 and 2006 are similar in sustained high NO in late November with a few
isolated peaks in late December (typically with light winds, strong surface inversions, and clear skies). It should be noted that
the 2003 peaks in NO in late November stand out as the highest in all four years. As noted earlier, this was also a period of
lowest cloud fraction and light winds. In 2000, the ozone hole filled in early November coincident with the breakup of the
polar stratospheric vortex: NO concentrations stayed very low through the entire observing period. In 1998, the ozone hole
filled briefly near the end of December followed by a very delayed (until the end of January 1999) formation of the thermal
tropopause due to a cooler than normal lowermost stratosphere.

### 3.2     Meteorological and other potential local and external influences on NO concentrations

Figures 1-4 revealed that there were well defined meteorological regimes that favored high NO concentrations that involved
wind directions aloft from the southeast, surface winds from the east, and a surface energy balance that was strongly
influenced by clear sky conditions. Other antecedents may include transport of $NO_X$ in the free troposphere that then mixes
to the surface supplying nitrate or alternatively, that air previously in contact with the surface was enriched from snowpack
emissions upwind of the station (Huey et al. 2004). As an example, Concordia Station in early December 2011(Berhanu and
Co-authors 2015) showed large increases in surface nitrate following a period of high wind in late November (Frey et al.
2015). Similarly at the South Pole in 2003 particularly high filterable nitrate was observed in the period JD328-338 (24
November to 4 December) as well as high methanesulfonate from 28 November to 2 December -- an indication of marine
sources (Arimoto et al. 2008). This period had the highest NO values recorded in the entire series of field programs (Davis et
al. 2008) similar to those found at Concordia in a similar time frame in 2011 (Frey et al. 2015). Furthermore, as suggested in



(Arimoto et al. 2008) air arriving at the South Pole may have had its origins in continental regions (due to much higher concentrations of the elements Pb, Sb, and Zn than in previous field programs) as well as in the ocean off Wilkes Land on the far side of the continent.

The conceptual description of Arimoto et al. depended on the December 2003 mean geopotential pattern which reflected a large high pressure region over east Antarctica and its large scale counter clockwise circulation. However, the pattern for the

ten days prior to December 4, 2003, when NO was the highest, was much more complex as shown in Fig. S12, with lower GPH over most of the continent and deep lows over the southern Weddell Sea, Enderby Land and Victoria Land all in position to provide transport of air from more northerly latitudes. Figure S13 shows the ERA-I trajectory for air at 700 hPa arriving near the South Pole at 12Z on 30 November 2003 from the vicinity of Wilkes Lake which extends between about $100^o$ E and $140^o$ E. The period after December 4 was consistent with the pattern described by (Arimoto et al. 2008) with

much higher GPH over East Antarctica. This leaves open the question of the origins of high snow nitrate in late winter/early spring insofar as October-early November is a period of vigorous transport from latitudes north of Antarctica following the maximum in the semi-annual oscillation and as evidenced by high cloud fraction (Neff 1999).

Episode "C" is quite interesting insofar as it is preceded by a period of high winds and a rapid surface wind shift from north to southeast following the winds at 300 hPa. Figure 7 shows this case in more detail using wind data from the South Pole

and AWS stations Henry (100 km along $0^o$ E) and Nico (100 km along $90^o$ E) as seen in Fig. 5. These data show that as the wind speed aloft drops, the surface wind speed also drops to less than 0.5 ms$^{-1}$ albeit sequentially first at Henry, then Nico, and finally at the South Pole. The increase in NO follows a surface wind direction shift and increase in speed from the SE that appears first at Henry and then four hours later at the South Pole. These shifts are quite abrupt whereas that at Nico is more gradual. The increase in NO at the South Pole does not coincide with the drop in wind speed to 0.5 ms$^{-1}$ (which would

have implied local accumulation or local sources) but rather follows during an increase in wind speed with the wind direction shift to the SE (from the basin shown in Fig. 5.) This suggests a hypothesis that this basin to the SE is an accumulation area for high NO that is exhausted after an hour or two (see Fig.7: with wind speeds of 2-4 ms$^{-1}$, transport distances of 7 to 14 km per hour might be inferred).

Tracing the progression of the event through the AWS and the South Pole, suggests that the event is triggered by a mesoscale

disturbance propagating from the NE that leads to boundary layer changes. However, referring to Fig. 7d, it appears that clearing skies quickly result in a low-level radiation inversion and associated shallow BL in which NO can accumulate. The 22-m tower data show that most of the radiative cooling is felt below 10 m. Of interest in this case is the greater contribution of mesoscale processes relative to the potential influence of katabatic forcing. Figure 8 shows geopotential and wind field at 650 hPa from ERA-I (~300-400 m above the surface at the SP) during this event at 12Z, JD 337 (3 December) between $90^o$ S

and $75^o$ S. In this case a small mesolow formed to the west of the SP and migrated to just north of the SP over 24 hours. It then intensified further north bringing air over longer distances from the E to NE back to the station but with a reduction of NO to about 200 pptv over the following day. The rawinsonde at 1048Z on Day 337 recorded winds from the NE at 8 ms$^{-1}$ below 100 m. By 2120Z the rawinsonde showed winds increasing in the lowest 100 m approaching 10 ms$^{-1}$. It is instructive to compare these winds to those estimated from katabatic arguments. An estimate of the katabatic acceleration can be

obtained from: acceleration = 9.8 ms$^{-2}$ $*$ $\Delta$T/T $*$ sin($\alpha$). With $\Delta$T=2C, T=250C, and terrain slope, sin($\alpha$)=0.002 (from Fig. 5),



the acceleration is ~0.5 ms$^{-1}$ per hour. Four hours at this acceleration rate would lead to a wind speed of 2 ms$^{-1}$ and a transport distance of several tens of km. In this case katabatic effects can account for only a small portion of the local circulation and Fig. 8 shows the dominance of meso-to-synoptic scale eddies in producing transport across the ice sheet. In particular, the small meso-low moved clockwise around SP until at 18Z it was in position to accelerate easterly flow at the surface. With

NO decreasing to 200 pptv following the initial peak of 800 pptv in the prolonged NE flow, one might reasonably assume that the initial extreme in this case resulted from short-term boundary layer effects but that prolonged levels of 200 pptv are fetch related and reflect accumulation upstream or transport from longer distances.

This case also highlights a unique feature of the South Pole, namely the low sun angle through the summer that allows a net loss of radiation from the surface in non-cloudy conditions (Carroll 1982). This is unlike the situation at sites at 75$^o$S and

80$^o$S where a convective boundary forms during the "daytime" such as Concordia Station (King et al. 2006) or Dome Argus (Bonner 2015). The above example of the role of the rapid onset of surface radiative cooling as clouds clear is analogous to observations from Concordia Station (Frey et al. 2015), which is subject to a strong diurnal cycle and show a similar rapid collapse in BLD in the late afternoon as the surface cools through long-wave radiation. Even though the actinic flux decreases dramatically relative to its value at local noon at Concordia, the BLD collapses even more rapidly as the production

of NO into the BL continues, leading to increases in NO to 0.3 to 1.0 ppbv (Frey et al. 2015). It was also concluded by Frey et al. (2015) that the rapid increase in NO with the collapse of the boundary layer in the early evening suggests that nonlinear HO$_X$-NO$_X$ chemistry need not be invoked in these cases. The case in Fig. 7 suggests that smaller enhancements of NO (say, less than 200 pptv) occur with longer fetches when NO can accumulate but large excursions such as those in Fig. 7 respond to rapid changes in the surface energy budget that lead to very shallow inversion layers. The subsequent evolution of NO

would then depend on chemical processes as well as changes in boundary layer mixing, such as those due to increased wind speeds. Given the apparent sensitivity to surface cooling under clear sky conditions, Fig. 9 shows the relationship between the magnitude of direct solar irradiance, NO, and surface wind direction. In Fig. 9a, NO>400 pptv only occurs when skies are essentially cloud-free whereas in Fig. 9b, wind directions > 45 degrees are associated with clear skies. Conversely, more persistent cloudy conditions and surface winds from the west to NNW are associated with lower NO. The few outliers for

wind azimuths >45$^o$ indicate the presence of occasional scattered clouds (in hourly data).

### 3.3    BLD calculated for 1998, 2000, 2003, and 2006 data

Given the success of the multiple linear regression technique described in Appendix A which used 22-m meteorological tower data with 2003 sodar data to develop regression equations which were then tested against independent data from 1993, we computed hourly BLD for each of the four years during which NO measurements were recorded. The main time period addressed by these estimates was from mid-November (or somewhat later if data collection started later) to the end of

December (for consistency). The next step was to test these data for the inverse relationship between NO and BLD as was found for 2003 (Davis et al. 2004a; Neff et al. 2008). If one bins the data to average out short-term fluctuations (Davis et al. 2004a; Neff et al. 2008) one finds a close inverse relation as shown in Fig. 10. Here we averaged BLD for 100 pptv bins of NO. Earlier results (Davis et al. 2004a; Oncley et al. 2004) applied empirical surface-layer similarity relations to observations of turbulence scales in the surface-flux layer to determine BLDs in the 2000 data set. In their results they found



BLD between ~80m and ~500m (Davis et al. 2004a) when they binned NO and BLD in sequential 30-point averages. In Fig. 10, for 2000, our binned BLD data fell between 28 m and 100 m, a factor of ~3 less in total range.

A key argument for high NO at the SP has been the presence of shallow boundary layers which Fig. 10 confirms. However, in contrast, direct measurements of BLD at Summit Station Greenland (3216m ASL, 72.6°N, 38.5°W) show values of NO in
435 the range of only 10-30 pptv with BLD on the order of 5 m (Van Dam et al. 2013). One potential explanation for this difference is the high snow accumulation rate in the summer at Summit Station of 5.0 to 7.5 cm per month in the summer, maximizing in July (Dibb and Fahnestock 2004, Castellani et al. 2015).

Results from 2003 stand out in Fig. 10. Recapping the findings of Neff et al. (2008), an unusual period of high NO and/or deeper BLD occurred from 24 November to 4 December. This period was unusual in that strong winds from the east
440 extended from the boundary layer to several kilometers aloft. As described in (Neff et al. 2008) the boundary layer was well mixed up to a capping inversion at about 100 m after an initial shallow phase. This was also a period of high methanesulfonate (MSA) and filterable nitrate that were likely of marine origin either from the Weddell Sea or from across the high eastern ice sheet from Wilkes Land and/or continental regions (Arimoto et al. 2008). During this period there were three bursts each lasting 30 to 50 hours of easterly wind and decreasing levels of NO with each burst (765 pptv, 512 pptv,
445 290 pptv, respectively). The first burst had NO>900 pptv coincident with high nitrate (0.5 μg m-3) and high MSA (0.046 μg m$^{-3}$). The highest MSA occurred toward the end of the first easterly burst (Arimoto et al. 2008).

## 4. Estimating seasonal $NO_x$ emission fluxes for the SP region

The development of a robust method for estimating BLD opens the door to estimating seasonal $NO_x$ emission fluxes. This is based on the assumption that the surface flux is in balance with the photochemical loss of $NO_x$ in the overlying BL column.
450 These flux estimates require several factors to be considered. First, the total BL $NO_x$ must be extrapolated from the near-surface measurement of NO. While only NO was measured in previous years, the 2006 measurements included both NO and $NO_2$. Based on those measurements, $NO/NO_x$ was found to be 0.65 +/- 0.08. This is consistent with $NO_x$ estimates derived from model calculations in previous years (Chen et al. 2001; Chen et al. 2004). Those calculations also indicate that the $NO_x$ lifetime exhibits nonlinear behavior increasing from ~7.5 hours when $NO_x$ is less than 200 pptv to more than 20 hours when
455 $NO_x$ exceeds 600 pptv (Davis et al. 2004a). Along with these effects, tethered balloon measurements SP during ANTCI 2003 show that the exponential roll off in NO concentrations from the surface to the top of the boundary layer can often be as much as a factor of five for very shallow boundary layers (Helmig et al. 2008; Neff et al. 2008). Using these vertical profiles, the BL column $NO_x$ abundance can be derived from surface NO observations. Given that the roll off in NO may not always be as severe as a factor of five, these are considered to be conservative estimates of the total BL $NO_x$.

460 Seasonal $NO_x$ emission fluxes are generated by taking daily average values of NO measurements and BLD estimates (based on Appendix A). Then based on the assumptions listed above, a total BL $NO_x$ and associate lifetime are estimated. The flux is then equivalent to the total BL $NO_x$ divided by the lifetime. It is important to do this calculation daily since changes in BLD can cause total BL $NO_x$ and lifetime to vary in different ways. Results for the four seasons as summarized in Table 1 are 12.5, 6.3, 13.7, and $9.1×10^8$ molec cm$^{-2}$ s$^{-1}$ for 1998, 2000, 2003, and 2006, respectively, assuming a fall-off factor of
465 five. These values are commensurate with expectations from Figure 10, with the highest seasonal flux in 2003 and the lowest





in 2000.   As can be seen in Fig. 11, NO concentrations at $BLD^{-1}$=0.03 $m^{-1}$ (from Fig. 10) scale almost linearly with the seasonal average flux ($r^2$=0.7).

Each year shows similar variability in daily flux estimates with standard deviations of 0.4-0.5 ($10^8$ molec $cm^{-2}$ $s^{-1}$). These estimates are generally consistent with other published flux estimates for the Antarctic plateau. (Oncley et al. 2004) estimated
a flux of $3.9\times10^8$ molec $cm^{-2}$ $s^{-1}$ at SP in 2000 over several days at the end of November 1998, which is essentially in agreement with the seasonal estimate for this work. Likewise, flux observations at Dome C have been found to range from $8.2\times10^8$ to molec $cm^{-2}$ $s^{-1}$ in 2009-2010 to $1.6\times10^9$ molec $cm^{-2}$ $s^{-1}$ in 2011-2012 (Frey et al. 2015; Frey et al. 2013). These larger values also exhibit a larger range of temporal variability due to the diurnal cycle at Dome C.

The range of seasonal flux values and daily variability is smaller than might be expected based on Fig. 10, but it is important
to note that under shallow BL conditions, nonlinear growth in $NO_x$ abundance can occur as radical concentrations become suppressed and $NO_x$ lifetime increases (Davis et al. 2004b). Nevertheless, the differences between years are significant and point to other potential factors affecting the seasonal $NO_x$ emission flux. One factor that has already been discussed at some length is the interannual difference in the breakup of the polar stratospheric vortex and associated ozone hole. With regard to nitrate photolysis, this is particularly important to determining the available UV actinic flux in the early Austral spring over
the Antarctic plateau. As shown in Fig. S10, this breakup happened much earlier in 2000, the year with the smallest seasonal flux. A second related factor has to do with the potential for recycling of $NO_x$ between the firn and atmosphere. As described in the following paragraph, secondary $NO_x$ emissions due to recycling are likely to grow in importance throughout the season and are likely influenced by primary emissions in the early spring.

The $NO_x$ emissions flux represents contributions from both primary and secondary release of nitrogen from the snow.
Primary release represents the photolysis and ventilation of $NO_x$ from nitrate initially deposited through snowfall. This nitrate can be much harder to release due to cage effects (Noyes 1956) associated with incorporation into the bulk ice phase. By contrast, secondary release of redeposited nitrate should recycle through the atmosphere much faster due to several factors. First, the redeposited nitrate is at the snow surface where higher actinic fluxes are available. Second, being adsorbed to the surface of ice crystals, redeposited nitrate should be more easily photolyzed.  In a laboratory experiment by (Marcotte et al.
2015), photolysis of nitrate on the ice surface was enhanced by a factor of three over nitrate in the bulk ice phase. Finally, redeposition of reactive nitrogen is in the form of both nitric and pernitric acid (Slusher et al. 2002).  Pernitric acid would be even more readily photolysed than nitric acid and its relative importance would also increase at lower temperatures. While representing these effects with any realism is beyond the scope of this analysis, more detailed modeling and representation of the factors influence $NO_x$ emissions are being investigated by other groups such as in (Bock et al. 2016; Frey et al. 2015).
Future laboratory and field studies are needed to further illustrate the complex mechanisms controlling snow $NO_x$ emissions and the chemistry above snow surface.

### 5.     Conclusions

We have described the influence of the synoptic- to-mesoscale weather patterns and their seasonal cycle on stable boundary layer characteristics at the South Pole in the spring-summer period that set the stage for high NO episodes on the Antarctic
plateau.  These included 1) the relative unimportance of katabatic forcing compared to the accelerations due to synoptic and mesoscale scale pressure gradients from November through January, 2) the effect of clearing skies locally that lead to





radiative losses and the rapid formation of very shallow inversion/boundary layers and high NO, and 3) the three-phase transition in spring for 300-hPa winds over the SP. These three phases corresponded to 1) a winter regime of transport of moisture over west Antarctica to the interior when the circumpolar trough is at its maximum and opens the possibility for the

transport of NO precursors from northerly latitudes, 2) a bimodal regime with 300 hPa winds alternating between northwest and southeast during November and early December as part of the seasonal cycle, followed by 3) a summer regime favoring 300-hPa winds from the Weddell Sea and warmer cloudy conditions. During the second phase, 300-hPa winds from the southeast favored clear skies, light surface winds and shallow inversions conducive to high NO concentrations at the same time the total column ozone was still low allowing higher actinic fluxes. 300-hPa winds from the northwest favored warm air

advection and cloudy conditions resulting in deep boundary layers and low NO concentrations. Because these boundary layer characteristics follow the same seasonal cycle as temperature, it is difficult to separate out the temperature dependence of nitrate photolysis rates from December into January.

We examined the frequency of occurrence of SE winds at 300-hPa and at the surface during late spring (Days 310-340) from 1961 to 2013 and found strong interannual variability. At 300 hPa the variance was significant but modulated on decadal

time scales with a maximum in the period 1995-2010. At the surface, the variance doubled with an upward trend of 30% in the mean in the last 30 years. We also found that extremes in the frequency of occurrence of SE winds at 300 hPa correlated well with extremes in easterly surface winds at the surface ($r^2$=0.4). This suggests that continuous multi-year monitoring may be advisable to avoid sampling bias. We also examined wind distributions aloft between the periods 1961-1980 and 1990-2010 and found no systematic change except that related to the weakening of the tropopause and increased coupling of winds

between 300- and 200-hPa. We concluded that the delay in the breakup of the ozone hole in recent decades has allowed higher actinic fluxes to coincide with the optimum time in spring when boundary layer conditions are conducive to high NO concentrations.

We also examined NO data collected in 2006 confirming the systematic relationship between NO concentrations and wind direction shifts to easterly with lower temperatures, lower wind speeds, and shallow boundary layers. In comparing 2006-07

with other years, we found similar frequencies of easterly surface winds in 2003, fewer in 1998, and the lowest in 2000. We carried out a case study from 2006 that occurred just prior to the final stratospheric warming. It was at this time that a minimum in cloudiness occurred, a minimum that is persistent in the long-term climatology over 50 years although subject to some synoptic "noise." Over a few hours in this case study, winds aloft rotated 180° from northerly to southerly (in a grid sense), a mesoscale cool front moved from the northeast (observed in two AWS stations 100 km from the Pole and captured

by ERA-I reanalysis) and skies cleared. With a clearing sky, a surface inversion formed below 10 m and NO climbed in 10-min averages of up to 1200 pptv with light surface winds from the southeast. NO>400pptv lasted only eight hours in this case with sustained moderate NO (150-400 pptv) lasting for another 48 hours in a sustained northeasterly flow (see Fig. 8). We argued that the source area for extreme NO concentrations was a shallow basin that extended about 100 km from the South Pole to the southeast. In this case we hypothesize that the short-term peak in NO occurred, not from transport in a

long-range katabatic circulation but rather originated from local, subtle terrain accumulation areas in the vicinity of the South Pole. This was coupled with rapid radiative losses from the snow surface that created a strong surface inversion below 10 m. In some respects, this situation is similar to the peaks in NO that occur at more northerly sites such as Concordia Station where an evening peak in NO occurs with the collapse of the daytime convective boundary layer into a shallow stable



boundary layer. In such cases long fetches are not necessary to achieve extremes in the concentrations of NO. However, in 2006, only 15% of the hourly NO data exceeded 400 pptv whereas 53% lay between 100 and 200 pptv. It is in this latter range that accumulation in long fetches associated with stable boundary layers and meso- and synoptic-scale circulations may be most relevant. These results suggest further studies of chemical and boundary layer processes along pathways more complicated than predicted by katabatic flow arguments, particularly with the subtle topography to the east and southeast of the SP.

We developed a method to estimate boundary layer depth using 20-m meteorological tower data together with direct depth measurements in 2003 and tested against an independent data set collected in 1993. Step-wise linear regression using wind speed and direction, temperature and the 2-to-20 m tower temperature difference produced regression equations that accounted for about 70% of the variance with direct observations. These equations were applied to all four seasons of NO data and confirmed the inverse depth relationships found in earlier studies. However, we also found significant year to year

differences in magnitude of these inverse relationships that suggested interannual variability in the surface fluxes of NO. To calculate $NO_x$ fluxes, the BLD data were used to obtain an estimate of the BL column $NO_x$ abundance (based on 2006 data where $NO/NO_x$ ~0.7). The abundance was derived from surface NO observations using a range of values of fall-off with height obtained from tethersonde profiles in 2003 and calculated using the increased lifetime of $NO_X$ for very shallow boundary layers. For each season, we found the average NO (scaled at 33.3 m from a least-squares fitting of NO versus

$BLD^{-1}$ for each of four seasons) which followed a nearly linear relationship with the seasonally averaged fluxes of $NO_X$ confirming the consistency of our analysis. The source of the year-to-year variability in fluxes was not fully resolved but the variability was consistent with similar results at Concordia Station.

Acknowledgements: NOAA/ESRL Global Monitoring Division personnel were responsible for SP data collection during the year 2006 starting on February 5 through January 15, 2007. NSF's Office of Polar Programs Grants OPP-9725465 and OPP-0230246 (PI D.D. Davis) provided partial funding for the ISCAT and ANTCI projects. The sodar data obtained in 1993 were supported by NSF Grant OPP 91-1896 (PI W.D. Neff).





**Appendix A Boundary Layer Depth Estimation**

In the main sections of the paper we discussed the large- and meso-scale influences on local micrometeorology and chemistry that can set the stage for enhanced levels of NO. Past work has implicated shallow boundary layers as a key ingredient in high NO episodes (Davis et al. 2004a). These shallow boundary layers were associated with lower wind speed, changes in wind direction, colder temperatures and greater static stability as indicated by the temperature difference between 2 and 22 m on a nearby tower (Fig. 4). Although a shallow BLD was proposed as a major factor underlying high-NO episodes, the only

direct BLD measurements were those carried out in 2003 using a sodar in combination with surface turbulence measurements and balloon profiling (Neff et al. 2008). However, such supporting measurements were not available for other field seasons.

    With this background, we examined regression analysis of NO and BLD against the various meteorological variables available to us for all four field seasons including the more recent data from 2006. This establishes some of the basic relationships between NO and BLD for the routine meteorological variables. However, because potential collinearity

between a number of the variables, we also pursued step-wise linear regression to find a subset of the variables that best account for NO and BLD for each observational period. This approach then identified a simple subset of variables to develop prediction equations for BLD through multiple linear regression (MLR). Establishing a relationship between NO and BLD, it then becomes possible to explore the unique chemical mechanisms which might be responsible for the high concentrations of NO at SP as compared with other sites and with the here-to-for under prediction by chemical models (Davis et al. 2008; Frey

et al. 2015).

    The most routine hourly variables available at SP include wind speed (WS) and direction (WD), temperature (T), and temperature difference across a 20-m tower ($\Delta T$). We also considered hourly direct solar irradiance (DR), net long-wave irradiance NIR, and solar zenith angle (SZA). (The variables DR, NIR, and SZA were available from www.esrl.noaa.gov/gmd.) Cloud fraction (CF) was available as a daily average from station climatological records whereas

bulk stability ($\Delta T_B$) was obtained from the twice daily rawinsonde in summer. Sodar data gathered in 1993 (Neff 1994) provided an additional test data set with which to evaluate the effectiveness of the 2003 MLR results. These data were stored as facsimile records rather than digitally [unlike the 2003 data where automated processing was used (Neff et al. 2008)]. Also the sodar in 1993 was operating in a Doppler wind measuring mode so the pulse resolution was coarser (~30 m) so echo layers and hence BLD appeared to extend higher by one half the pulse length or 15 m. To correct for this effect 15 m was

subtracted from the BLD estimates.

    Linear regression results are shown in Table A1 for NO in each of the four field seasons along with results for measured BLD given in Table A2 for 2003. In all four years, the dependence of NO on near surface stability $\Delta T$ is consistent. In 2003 and 2006, NO depended primarily on WD, T, and $\Delta T$; WS was more important in 2003 than 2006. In Table A2, we show overall regression results for BLD in 2003 as well as a breakdown into two periods corresponding to the dynamical changes around

JD340 (6 December). The early period, JD326-340, is dominated by WS, T, SZA, and net radiation. The change in the impact of bulk temperature difference in this period $\Delta T_B$, from the rawinsonde, marks the change due to the seasonal cycle as in Fig. S4a. However, based on Fig. S4a, katabatic forcing will be small compared to that due to large scale weather systems. During the later period, JD340-361, WS, WD, $\Delta T$, and cloud fraction CF play bigger roles.



In general, NO and 2-m temperature have robust seasonal trends from October to late November except in the year 2000 when the polar vortex broke up a month earlier than normal and the temperature showed no trend after early November and in 2003 when the trend persisted into early December. In 2003, the dependence of NO on SZA is quite clear with moderate dependence on direct solar irradiance whose variability depends to some degree on cloudiness (see Figs. 4 and 9: With respect to Fig. 9a, the simple linear regression between NO and DR in Table I does not capture the threshold effect evident in the figure. (Cases with high DR in Fig. 9a and low NO usually occur under high wind conditions with deep boundary

layers.) The dependence of BLD on tower-measured $\Delta T$ is greater than on the bulk inversion temperature difference $\Delta T_B$ which is typically measured over several hundred meters: This is consistent with the fact that the BLD is usually much shallower than the background temperature inversion and more consistent with the $\Delta T$ measured on the 22-m tower. When $\Delta T$ is dominant over $\Delta T_B$ it indicates that the shallow temperature inversion is likely due to surface radiative cooling (with clearing skies) rather than synoptic weather changes. Comparison of these correlations with those from the more extended

data in 1993 sheds some insight into the appropriate variables to use in our MLR analyses (Table III).

From Table A3 (1993 Data), the dependence of BLD on wind speed increases significantly from October to December while the dependence on $\Delta T$ is remarkably consistent; after October, there is little dependence on either cloud fraction or bulk inversion strength. The increased dependence on WS in December is consistent with changes following the breakup of the polar vortex and the progression of the seasonal cycle. Comparing monthly averaged $\Delta T$ with $\Delta T_B$ yields the

result: October (1.44/16.5 $^o$C), November (0.34/8.3 $^o$C), and December (0.47/0.10 $^o$C). Typical bulk inversion strength was obtained over average depths of 542m, 579m, and 622m for each month respectively. This demonstrates the strong seasonal cycle in inversion strength and the changing seasonal impact on katabatic forcing of surface winds (see Fig. S2a). In December the near-surface (2-22m) inversion is much stronger than the bulk value from the rawinsonde suggesting that any slope effects will reflect nearby terrain variations such as those shown in Fig. 5. It should be noted that the effect of WD on

BLD is greatest in the transition month of November. With a weaker inversion strength in the summer, those winds (December) respond mostly to meso- and synoptic scale weather influences such as cloudy versus clear skies and/or high versus low wind speeds (Neff 1999) rather than katabatic forcing.

**A.1    Linear regression**

Because of the potential covariance of meteorological variables (e.g., often when the wind direction shifts to the SE,

wind speed, $\Delta T$ and temperature all decrease: see Fig. 6), we used stepwise linear regression (using SPSS) of NO and BLD with principal meteorological data as summarized in Table A4. The value of stepwise regression is that it orders the results by magnitude of the importance of variables to the regression. In Table A4, step-wise regression reveals year-to-year variability in important variables depending on the year-to-year variability of dominant meteorological regimes. It should also be noted that the step-wise regression process can produce a different sequence of regression coefficients than those

obtained from individual regressions as can be seen in comparing Table A1 and Table A4. The process also excludes variables that are collinear with other variables in the time series. In this case, results with at least an improvement of 0.02 in $r^2$ are presented as shaded. For BLD in 2003 the principal predictors are WS (56%), T (10% improvement), and WD (7% improvement). Including $\Delta T$ adds little (0.01) because it is significantly correlated with both WS and WD ($r^2=0.3$ for both). When the stepwise regression is applied to only JD340-361, the ordering is WS, WD, and $\Delta T$ with T being dropped because

there is no temperature trend. When examining the results for NO, the year-to-year differences in meteorology can be seen in



the dominant independent variable for each year: 1998: WS, 2000: ΔT, 2003:T, 2006:WD.  For the most part, meteorology can account for about 50% of the variance in NO leaving unknowns in the emission source terms unaccounted for.  2003 was an exception with 78% of the variance for which meteorology provided the explanation for the variability in NO.

Given the dependence of BLD and NO on WS, WD, T, and ΔT for various periods we used these four variables to obtain regression equations for 2003 and 1993 using sodar derived BLD.  For testing purposes the regression equation from 2003 was applied to both 2003 and 1993 as well as the regression equation from 1993 to both 1993 and 2003. The equations used were:

2003: $BLD=66.9+18.8*WS+0.23*WD+2.9*T-16.0*\Delta T$ (A1)

1993: $BLD=68.3+16.1*WS-0.04*WD+2.3*T-10.0*\Delta T$ (A2)

For 2003, $r^2=0.74$ with all the variables significant at the 0.000 level; for 1993, $r^2=0.64$ with all the variables, except WD, significant at 0.000 (WD was at 0.080).  Shifting the time series 10 days earlier made all variables highly significant showing the importance of changing weather regimes each year and sensitivity to the length of the time series.  The major difference between 2003 and 1993 lies in the relative weights given to WD and ΔT: In 1993 they are of the same sign (reducing BLD) whereas in 2003 they are opposite in sign and partially cancelling. Figure A1 a,b,c shows the results of applying the MLR technique (using WS, WD, ΔT, and T) to 2003data while Fig. A1 d,e,f shows comparisons using 1993 data.  As can be seen in the figures the regression equations capture the transitions between shallow and more deeply mixing boundary layers quite nicely.  Because the minimum range for the sodar observations was about 15 m, we set all MLR estimated BLD values less than 15 m (some were negative) to that value.  For deeper observed BLD in 2003, the MLR for both 2003 and 1993 regression equations underestimate the depth; for shallow BLD, the MLR overestimates the depth. Overall the MLR method captures about 70% of the variance for both 2003 and 1993.  It is also of note that the 2003 MLR, based on observations only to 160 m, does capture the deeper observed BLD in 1993 (Fig. A1) suggesting that wind speed dominates the regression fit in those cases.  Figure 10c provides a scatterplot for 2003 data using 1993 and 2003 regression fits with $r^2=0.96$. For 1993, the data appear to have two modes perhaps because of a weather regime shift not captured by the MLR method using just a few meteorological variables (e.g., there was a major change in bulk inversion strength $\Delta T_B$ around mid-December).  In this case the two equations still agree quite well with $r^2=0.93$.

### A.2 Results

We have extended the past analysis of boundary-layer depth (BLD) effects on NO concentrations from data collected at SP  in 2003 to new data collected in 2006-07 as well as past field programs carried out in 1998 and 2000-2001. Stepwise linear regression showed that the principal variables affecting both NO and BLD were wind speed and direction, temperature, and low-level static stability as measured on a 22 m tower.  We then used these variables to develop a multiple linear regression equation for BLD using sodar data from 2003 and simple meteorological tower measurements.  When we tested the regression equation derived from 2003 data against independent sodar data collected in 1993, we accounted for 60-70 percent of the variance in estimating BLD.  We then applied these equations to each year of NO data to examine the NO response to BLD.  When the data were binned in 100 pptv bins (Fig. 10), we found a linear fit between NO and 1/BLD with regression fits accounting for 88% to 97% of the variance confirming past results.  When we looked at the binned data for

(c) Author(s) 2017. CC BY 4.0 License.




BLD=100 m, all four years converged with NO levels of about 50-100 pptv.  However, at BLD=25m there was a systematic
spread in NO according to observation year resulting in NO that ranged from  200 to  1000 pptv. In looking at the results of
step-wise linear regression for NO we found a different primary variable in the regression for each year: 1998: wind speed,
2000: $\Delta T$, 2003: T, and 2006: WD.  This  primarily reflects the different weather regimes for each year and the need for
multi-year observations to fully understand the various factors leading to high NO concentrations.

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




# Tables

Table 1 Assessment of ISCAT/ANTCI NO$_x$ flux at SP for BL concentration fall-off of factor of 5

| $Year^a$ Time-period | $10m\text{-}NO_x{}^b$ molec/cm$^3$ | $Avg\text{-}NO_x{}^c$ molec/cm$^3$ | $BLD_{avg}{}^d$. cm | $NO_xCD_{f5}{}^e$ molec/cm$^2$ | $\tau_{NOx}{}^f$ s | $Estimated$ $NO_x\ Flux^g$ molec/cm$^2$/s |
|---|---|---|---|---|---|---|
| 2006 11/15-12/31 | 6.8 x 10$^9$ | 4.7 x 10$^9$ | 6.1 x 10$^3$ | 2.9 x 10$^{13}$ | 3.2 x 10$^4$ | **9.06 x 10$^8$** |
| 2003 11/22-12/27 | 10 x 10$^9$ | 6.5 x 10$^9$ | 8.1 x10$^3$ | 5.2 x10$^{13}$ | 3.8 x 10$^4$ | **1.37 x 10$^9$** |
| 2000 11/15-12/31 | 3.2 x 10$^9$ | 2.1 x 10$^9$ | 8.3 x 10$^3$ | 1.7 x 10$^{13}$ | 2.7 x 10$^4$ | **6.30 x 10$^8$** |
| 1998 12/1-12/31 | 7.8 x 10$^9$ | 5.4 x 10$^9$ | 7.5 x 10$^3$ | 4 x 10$^{13}$ | 3.2 x 10$^4$ | **1.25 x 10$^9$** |

(a)Year of NO or NO$_x$ measurement and time period (Year-time period ; (b) average NO$_x$ or NO measured at 10m - NO converted to NO$_x$ as discussed in text ; (c) average NO$_x$ concentration within BLD; (d) average BLD ; (e) average NO$_x$ column density, given BL concentration fall-off factor of 5; (f) average NO$_x$ lifetime ;(g) Estimated

NOx flux – derived from average column density divided by the average NO$_x$ lifetime.

Table A1 Regression (r$^2$) of Meteorological Variables with NO. r$^2$>0.2 in bold. No trends were removed.

| | WS | WD | T | ΔT | CF | ΔT$_B$ | SZA | DR |
|---|---|---|---|---|---|---|---|---|
| **NO 1998 (JD 335-365)** | 0.15 | 0.07 | **0.26** | **0.26** | 0.15 | 0.00 | 0.11 | n/a |
| **NO 2000 (JD 320-366)** | 0.14 | 0.16 | 0.11 | **0.41** | 0.08 | 0.01 | 0.00 | n/a |
| **NO 2003 (JD 326-361)** | **0.29** | **0.45** | **0.51** | **0.25** | 0.18 | **0.34** | **0.37** | 0.14 |
| **NO 2006 (JD 320-379)** | 0.18 | **0.25** | **0.56** | **0.36** | 0.06 | **0.34** | 0.07 | 0.15 |






Table A2 Linear Regression of BLD with observed meteorological variables in 2003

|  | WS | WD | T | ΔT | CF | ΔT$_B$ | SZA | DR | NIR |
|---|---|---|---|---|---|---|---|---|---|
| **BLD 2003 (JD 326-361)** | **0.56** | 0.10 | **0.25** | **0.32** | 0.03 | 0.14 | 0.13 | 0.07 | 0.00 |
| **JD 326-340** | **0.48** | 0.01 | **0.73** | 0.15 | **0.21** | **0.23** | **0.72** | 0.02 | **0.58** |
| **JD 340-361** | **0.67** | **0.46** | 0.06 | **0.53** | **0.32** | 0.10 | 0.19 | 0.16 | 0.14 |

Table A3 Regression ($r^2$) of Meteorological Variables with BLD, October-December, 1993(no

radiation data in 1993)

|  | WS | WD | T | ΔT | CF | ΔT$_B$ |
|---|---|---|---|---|---|---|
| **OCT** | **0.33** | 0.07 | **0.20** | **0.23** | 0.02 | 0.17 |
| **NOV** | **0.37** | **0.20** | 0.02 | **0.27** | 0.06 | 0.00 |
| **DEC** | **0.64** | 0.01 | 0.15 | **0.24** | 0.01 | 0.00 |
| **JD 326-361** | **0.61** | 0.04 | 0.17 | **0.24** | 0.00 | 0.00 |



Table A4 Stepwise linear regression analysis ($r^2$) for BLD and NO (mid-November-December.)

Stepwise improvements of 0.02 or better are highlighted.

| | | | | | |
|---|---|---|---|---|---|
| **BLD** | **2003(all)** | **0.56** | **0.66** | **0.73** | 0.74 |
| | Variable | *WS* | *+T* | *+WD* | *+ΔT* |
| | **2003 (Days 340-361)** | **0.67** | **0.72** | **0.74** | Excluded |
| | Variable | **WS** | **+ΔT** | **+WD** | T |
| | **1993** | **0.60** | **0.63** | 0.64 | Excluded |
| | Variable | *WS* | *+T* | *+ΔT* | *WD* |
| **NO** | **1998** | **0.25** | **0.40** | **0.43** | **0.47** |
| | Variable | **WS** | **+ΔT** | **+T** | **+WD** |
| | **2000** | **0.42** | **0.51** | **0.54** | 0.55 |
| | Variable | **ΔT** | **+WS** | **+T** | +WD |
| | **2003** | **0.51** | **0.76** | **0.78** | 0.79 |
| | Variable | *T* | *+WD* | *+WS* | *+ΔT* |
| | **2006** | **0.32** | **0.48** | **0.53** | **0.59** |
| | Variable | *WD* | *+ΔT* | *+T* | *+WS* |




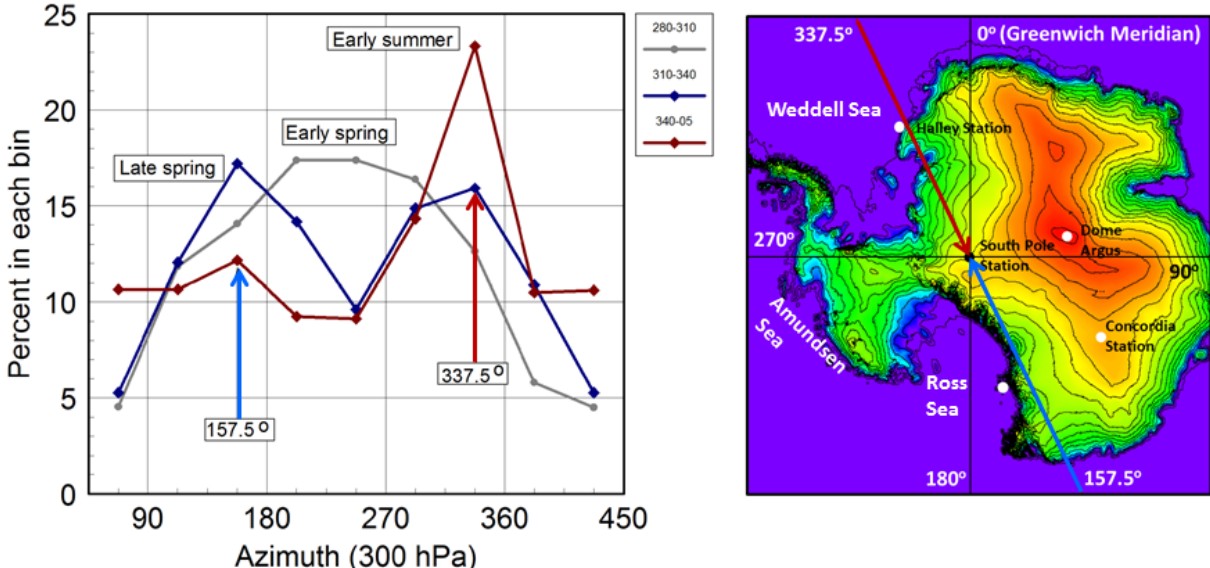

Figure 1. Left: Histograms of wind directions sorted in 45° bins (>5ms⁻¹) at 300 hPa for early spring

(Days 280-310, gray line) and late spring (Days 310-340) and early summer (Days 340-370) over the

period 1990-2010.  Right: Topographic map of Antarctica.  In late winter/early spring winds aloft are

primarily from the direction of West Antarctica and the Amundsen Sea (~270°).  In late spring, 300-hPa

winds become bimodal in direction, with higher probabilities from ~157.5° and ~337.5° and fewer from

the west.  Past work has indicated that winds from 157.5° bring cooler temperatures and lighter wind

speeds to the South Pole: this direction corresponds to an inland directed synoptic scale pressure

gradient.  Conversely, 300-hPa winds centered on 337.5° are typically associated with higher surface

winds, increased cloudiness, and warmer temperatures: this direction reflects an off-shore directed

synoptic pressure gradient.  In early summer, winds at 300 hPa increasingly favor a north-northwest

direction from the Weddell Sea.





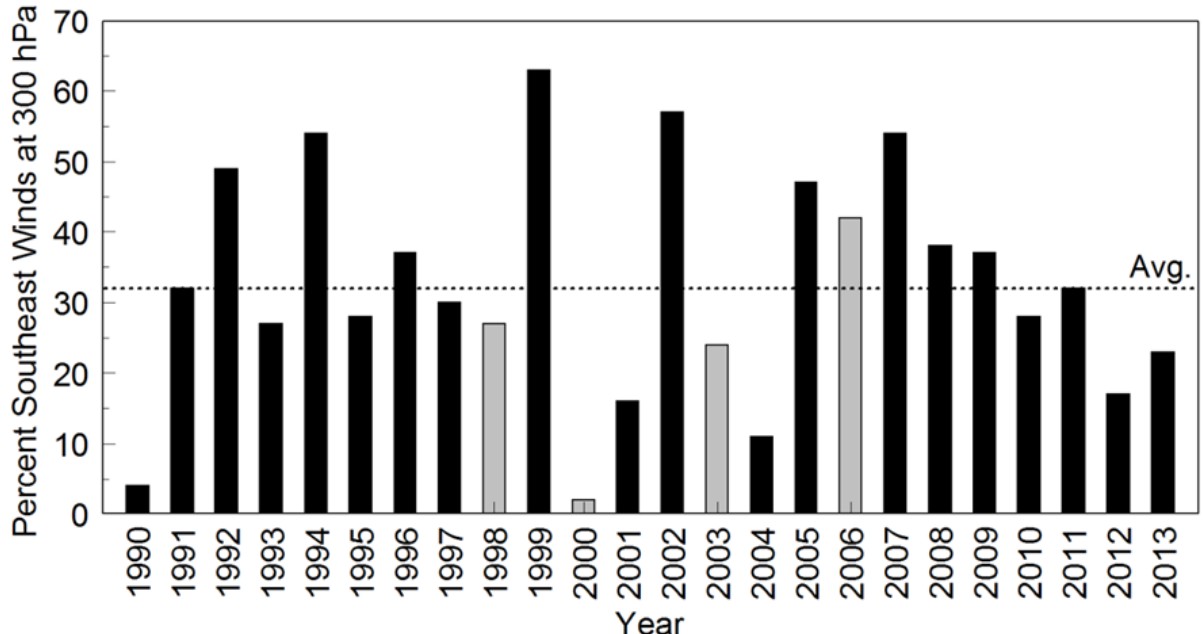

Figure 2. Percent of 300 hPa wind directions between $67.5^{\circ}$ and $202.5^{\circ}$ for each year 1990 to 2010 for mid-November through December. Experiment years are shaded gray. The average over 20 years is shown by the dashed line.






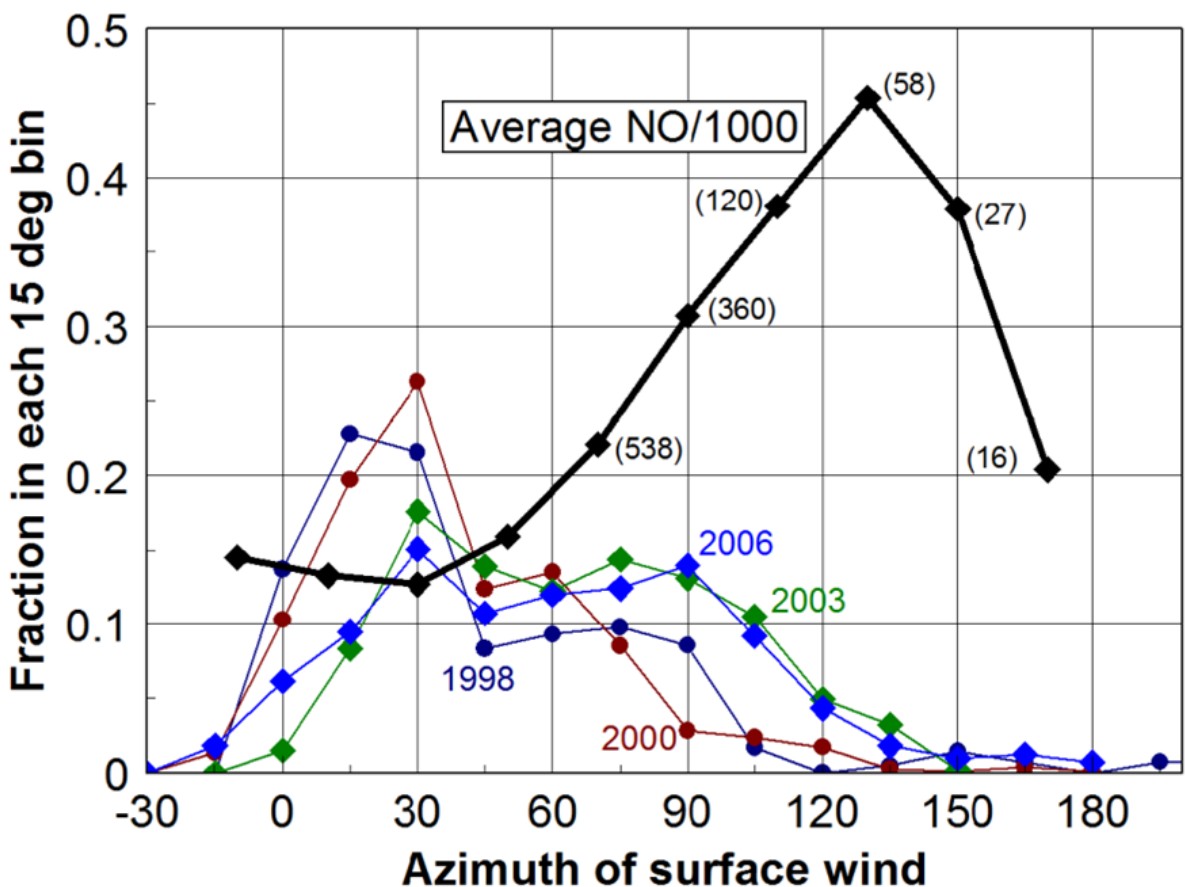

Figure 3. Comparison of surface wind direction distributions for 1998, 2000, 2003, and 2006 and the
average NO over four years as a function of wind azimuth. Wind directions between 210° and 340° (-
20°) are not included because of potential contamination. from the station and aircraft operations.   A
total of 3300 hourly values of NO were averaged in 20° wind-direction bins. Number included in each
average >200 pptv are indicated in (_). The largest NO occurs with surface wind directions aligned
roughly from the direction of Concordia Station (123°E, see Fig. 1).






**Figure 4.** (a) Time series of observer-derived cloud fraction averaged for 2000-2010 (red line) at the South Pole and the field season of 2006-07(black line) showing the consistency of individual years with the long-term average. A 14-day running average (red line) was used. The period of final warming of

the stratosphere in 2006 ($t_{200hPa}$-$t_{300hPa}$>0) and the period of rapid increase in total column ozone are indicated symbolically for JD 337-342. The date of the maximum in $j(NO_3^{-1})$ is indicated by an orange diamond (Day 336) from (Davis et al. 2010). (b) The expanded time series of cloud fraction for JD 316-350 showing relative minima on JD327 and JD337 as well as the timing for the breakdown of the ozone hole (as seen in $J_{NO3^-}$ nitrate photolysis rate coefficients scaled for plotting) and stratospheric warming.

(c) Expanded view of NO, Delta T (2-10 m), $J_{NO3^-}$ (scaled), and direct radiance time series for JD 335-340. Daily cloud fraction, derived from climatological observer records, is shown using green bars. Note the decrease in $J_{NO3^-}$ prior to the increase in NO.



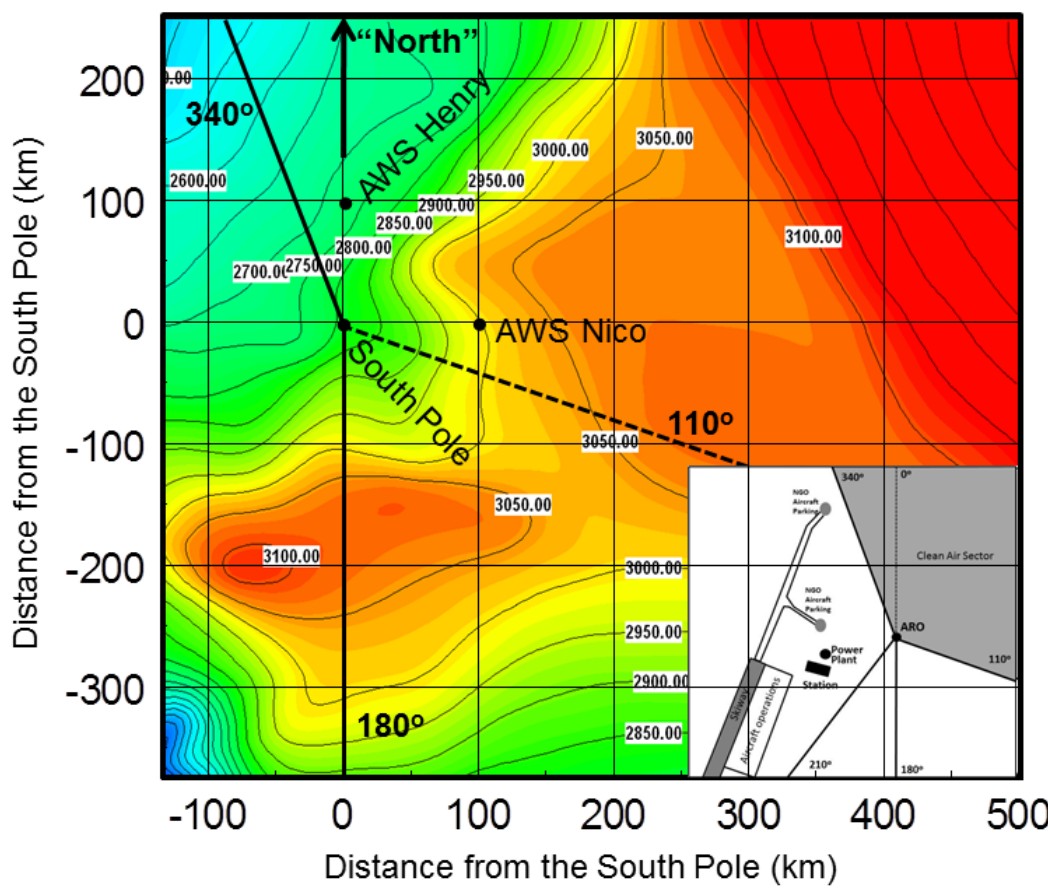


**Figure 5.** Local terrain around the South Pole and the location of station facilities and the clean air sector (insert). Contour interval is 50 m. The clean air sector extends from 340° to 110° whereas the major station activity is located between 340° and 210°. Note that the prevailing surface wind (northeasterly) is actually along terrain isopleths upstream of the station except for the last 50 km.

"Downslope winds" come from directions generally between east and south. Automatic Weather Stations (AWS) Nico and Henry which have been operating since 1993 are also indicated lying about 100 km from the South Pole. Highest NO is observed when the wind directions are from the direction of the terrain "gap" to the southeast of the Station.







**Figure 6.** Time series of NO, wind direction, speed and 2-m temperature for November 2006 through mid-January 2007 (Days 305-380). Data gaps at the end of December are due to either missing NO and/or meteorological data. Shaded areas and labels A-H are cases where NO >300 pptv. Case "C" is that highlighted in Fig. 4 which occurs at the time of stratospheric warming and total column ozone

increase. The wind azimuth of $60^o$ is also noted (corresponding to Fig. 3 for directions associated with higher NO). Periods of clear skies are indicated on an hourly basis by hash marks at -20 C and daily cloud fraction less than 6/10ths at -15 C on the temperature plot: Note the increase in temperature around JD 320 associated with cloudy skies, increase in wind speed, and shift in wind direction.



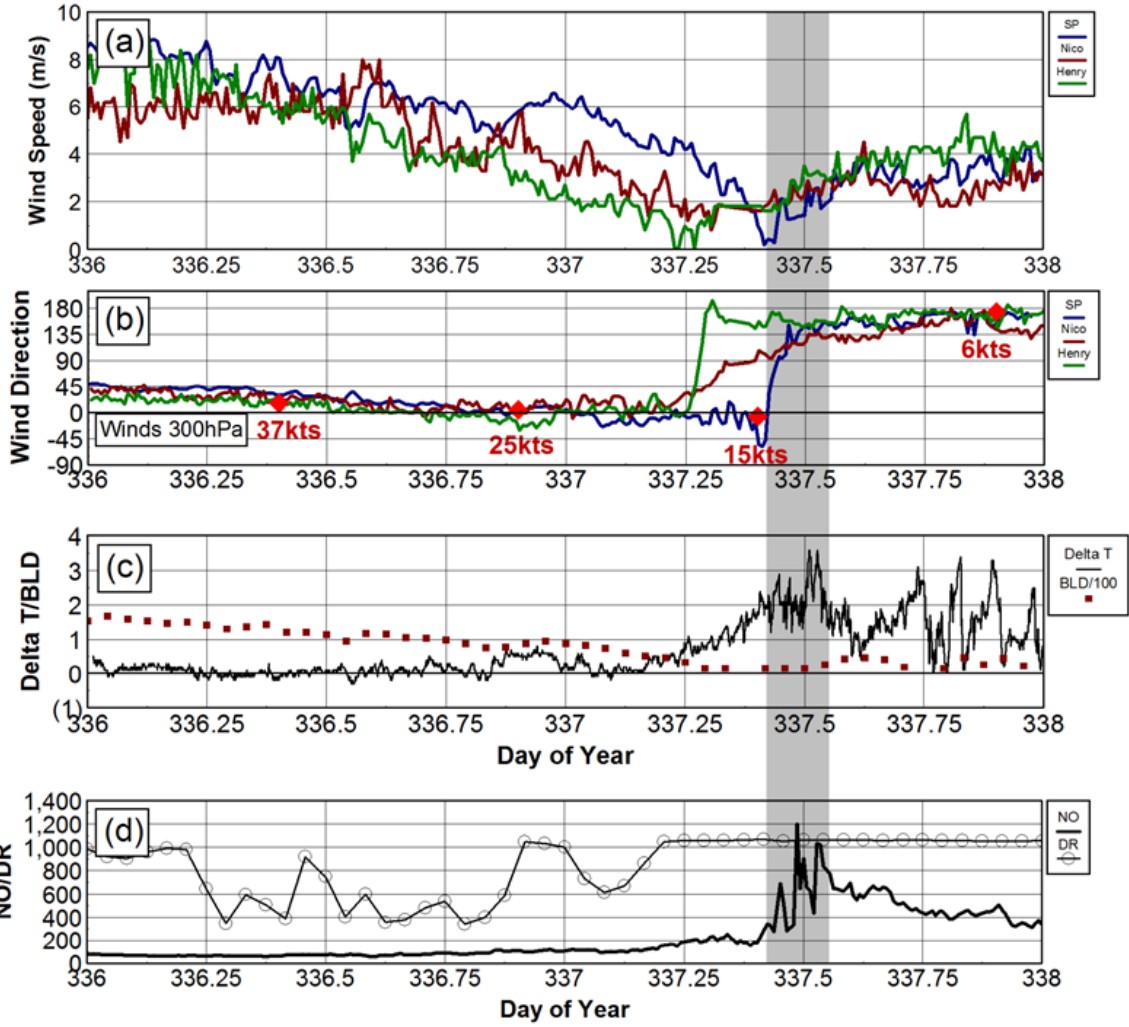

**Figure 7**. Time series of (a) Wind Speed, (b) Direction  (c) Delta T and BLD and (d) NO and direct

radiance (DR) for Days 336-338, 2006 (Case C in Figure 6).   In (b) we indicate wind speed and

direction at 300 hPa (red diamonds) at the actual launch times of the rawinsondes (2-3 hours prior to

reporting times of 00Z and 12Z).  Of note is the decrease in surface wind speeds from over 8 ms$^{-1}$

(sufficient for blowing snow) that follow the same trend as at 300 hPa.  We have also indicated the BLD

calculated later in Section 4.3 that shows the rapid decrease in BLD.




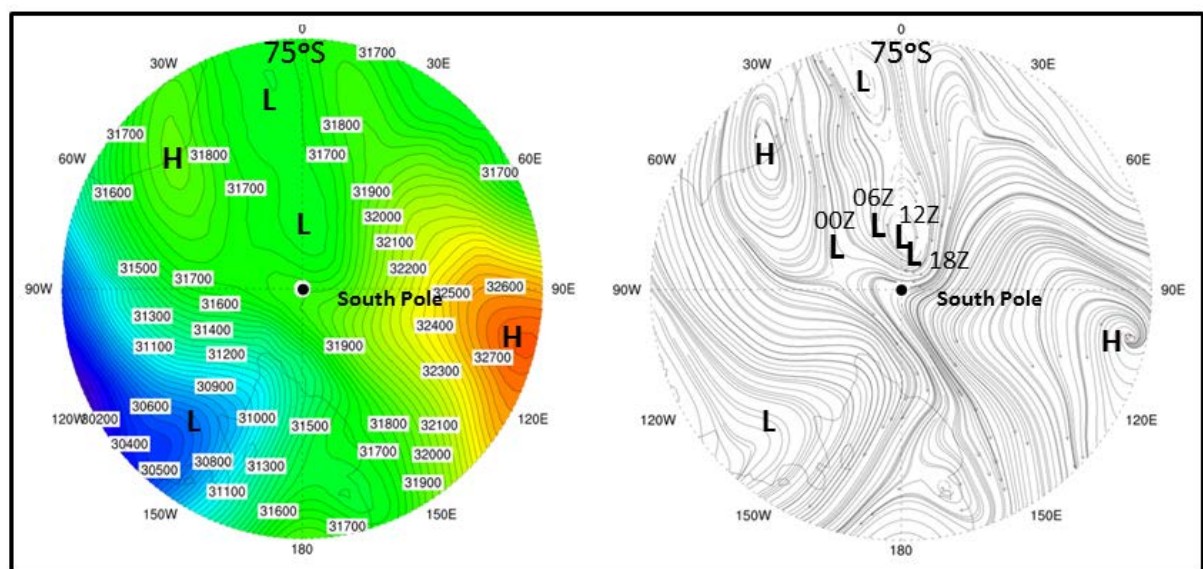

Figure 8. Geopotential at 650 hPa at 12Z on JD 337 (3 December 2006) and wind field (using curly vector representation) from ERA-I. A trough of low geopotential lies along $0^o$ with higher geopotential and anticyclonic winds over the high plateau. Lows and highs are indicated and the location of a small cyclonic circulation at 6-hr intervals on JD337 that brought easterly winds to SP. By 12Z on JD 338, the low to the NNE had intensified and produced a consistent NE winds to SP






Figure 9. (a) Direct radiance (DR) versus NO (hourly) for 2003 showing that values of NO>400 pptv
occur with virtually clear skies (DR>80% of the maximum). (b) Direct radiance versus surface wind
direction (hourly) for 2003 with average DR and number of observations in 45-deg bins, showing cloudy
to partly-cloudy conditions prevail with surface wind directions less than 45 degrees.






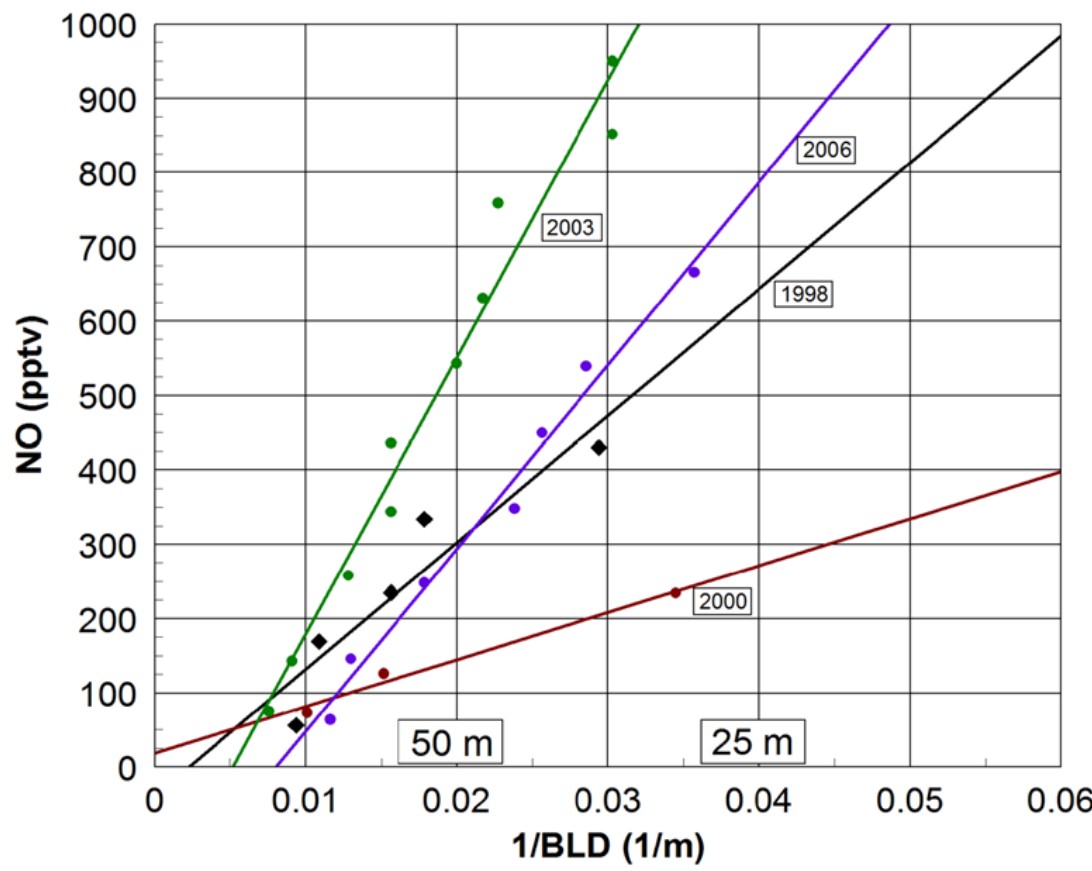

Figure 10. NO versus 1/BLD for BLD averaged for each 100 pptv bin of NO. $r^2$ is as follows: 1998: 0.88; 2000: 0.98; 2003:0.97; 2006: 0.98.



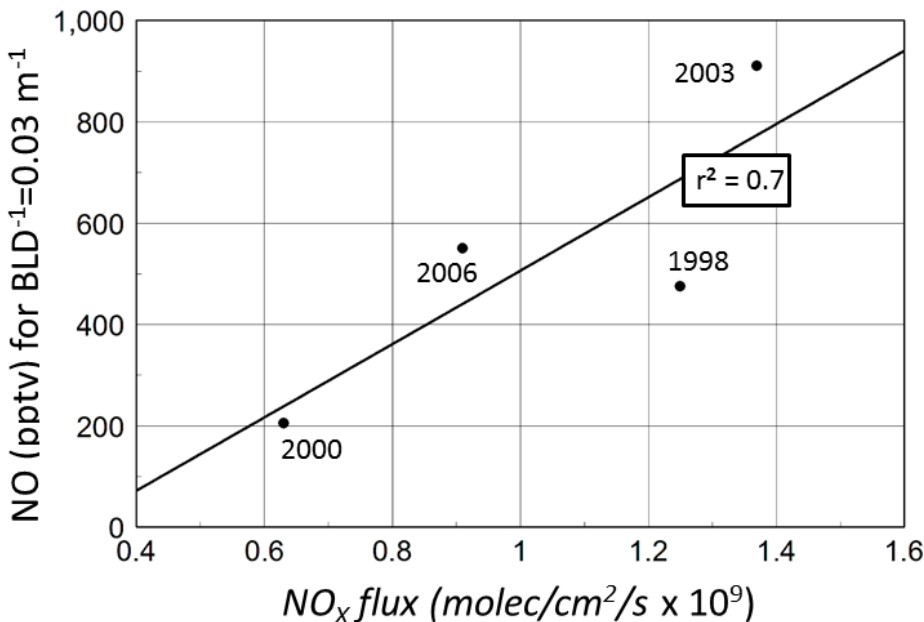


Figure 11. NO (BLD = 33.3m) versus seasonally averaged $NO_X$ flux from Table 1 using a fall-off factor of five (more typical of shallow boundary layers).



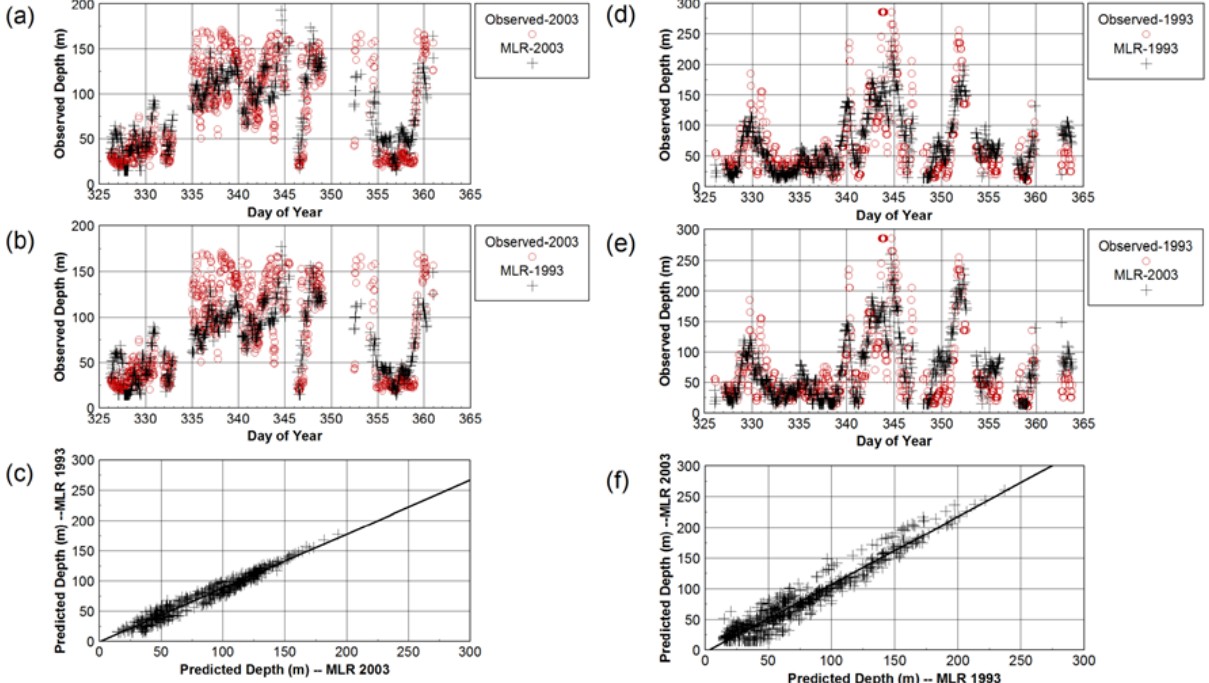

Figure A1. a) Comparison of 2003 observed BLD with that predicted by the 2003 MLR equation with $r^2$=0.71, b) Comparison of 2003 observed BLD with that predicted by the 1993 MLR equation with $r^2$=0.67, c) least-squares fit for MLR1993 versus MLR 2003with $r^2$=0.96, d) Comparison of 1993observed BLD with that predicted by the 1993 MLR equation with $r^2$=0.65, e) Comparison of 1993 observed BLD with that predicted by the 2003 MLR equation with $r^2$=0.61, f) least-squares fit for MLR2003 versus MLR 1993 with $r^2$=0.93.