# Peer review of "The Meteorology and Chemistry of High Nitrogen Oxide Concentrations in the Stable Boundary Layer at the South Pole"

_Atmospheric Chemistry and Physics, 2017_

## Referee Comment (RC1) · Anonymous Referee #1 · 17 Oct 2017

Neff et al. SP NOx

This manuscript focuses on the factors (mainly dynamical, with some attention to chemical processes) that contribute to large, but variable, mixing ratios of NO at South Pole (SP) during the spring to summer transition. Previously unpublished measurements of NO and NO2 during the 2006-2007 field season are presented and discussed in context of observations made in 3 earlier campaigns (1998/99, 2000/01, and 2003/04 seasons). Several previous papers by this team have established that the surprisingly high levels of NO sometimes seen at SP are sustained by snow to air flux of NO produced by photolysis of nitrate in the snow, building up when low boundary layer height

(BLH) limits vertical mixing. Low BLH resulted from low wind speeds and clear skies, with the latter also enhancing the rate of nitrate photolysis. Intervals with low BLH generally coincided with easterly winds, leading to a suggestion that some of the NO during extreme events was transported down slope to SP, with extensive recycling of redeposited nitric and pernitric acids combined with non-linear NOx chemistry resulting in mixing ratios approaching and even exceeding 1 ppbv.

The addition of the 2006 results confirms earlier findings but does not produce any fundamental new understanding. However, this manuscript does extend prior papers by exploring the links between BL dynamics at SP and synoptic and mesoscale dynamics in the upper troposphere and even into the stratosphere over Antarctica. In my opinion the key findings are that high NO at SP are most likely to occur when the flow at 300 hPa is from SE, which is found to correlate with relatively weak surface winds from the E-S quadrant and generally clear skies, factors leading to low BLH. It is also pointed out that the south-easterly flow at the surface is due to large scale pressure gradients, and not katabatic flow as had earlier been suggested.

These findings are significant enough to merit publication, but I find the manuscript overly long, repetitive at times, and hence not as accessible as it could be. In particular, I find that a lot (too much) of the discussion relies on figures in the supplemental material, and that graphical or tabular comparisons of NO in the 4 different seasons are never provided in the main body of the manuscript. Figure S10 does provide such an overview, but I suggest it would work better as a 4 panel plot (one panel for each season) and should be moved out of the supplement to someplace early in the main text. Current version of sections 2.1 and 2.2 go into great detail about meteorological similarities and differences across the 4 sampling seasons, but rarely relate these to observed NO concentrations at SP (which seems like it should be the main point of paper). Referring (probably often) to a revised figure S10 put into section 2 would help. Similarly, Fig 2 would be improved by adding some kind of NO metric (e.g., season mean, or fraction of sampling hours > than some threshold (250 pptv is used

elsewhere)), and the heavy black line showing average NO as function of surface wind direction in Fig 3 now averages all 4 seasons, unnecessarily obscuring interannual variations which are important and interesting. In the following list of detailed comments I mix purely editorial suggestions (typos, etc.) with comments related to streamlining the narrative by reducing redundancy and interesting side stories. Comments are linked to line numbers in the on-line pdf, hope these are conserved across different platforms.

69-73 Would help to indicate the length of these weather cycles. Seems the timescales are a few days of clouds, then a few with clear skies and strong inversion based on case studies presented much later, but here the reader could assume you mean seasonal changes.

73 I would argue that high albedo and low SZA are characteristic of most of Antarctic plateau. What is "unique" about SP is that the SZA barely changes within 24 hour day. Later on it seems the SZA is felt to be significantly lower at SP than at 75 or 80 S, but is that true averaged over a full day? Might help just to be quantitative about meaning of "low" SZA.

78 "omnipresent" seems an odd description of the inversion, given you just said the really stable boundary layer reforms every weather cycle.

87-88 words missing here, like "Antarctica" or the Antarctic ice sheet. Does not make sense to describe storm track as pole centric, or covering so many

95-96 Does not seem the semi-annual oscillation or the circumpolar trough are ever mentioned again, so why introduce the terms here? Are they controlling NO at SP?

98-100 Another interesting tidbit about large scale flow in winter that seems to have nothing to do with NO at SP in Spring/Summer

103-104 Is it fair to compare one 135 degree sector to another that is only 90 degrees? I also note that if wind direction was completely uniformly distributed the larger sector would account for 37.5% and the W to N quadrant 25%, making the quoted 43 and 31
**Interactive comment**

% fractions observed only slightly enhanced. Figure 1 makes the point, so consider deleting this sentence.

109-113 These last 2 sentences repeat ideas introduced earlier (in this paragraph, also in first paragraph of section 2, and in earlier papers). It is also odd to be making predictive statements about how NO should respond, rather than point to data that confirms the points discussed. Granted, the 2006 results have not yet been show, but the other 3 campaigns are published.

114-121 Perhaps unnecessary detail, the correlation is interesting and useful for chemists on its own.

121 can, however, vary—-→ varies

124 mention 1999 and 2002 and say they coincide with a single year with sudden strat warming. Which one had the warming?

125 Any decadal periodicity is not at all obvious in Fig 2. Even if it is there, is it important to understanding any differences between the 4 years (spread over 9 years) with observations at SP.

129-130 Is it really a new finding that NO at SP is not tightly controlled by any single parameter? Seems this has been noted in earlier papers.

134-170 A little confusing here, with first sentence saying SE winds aloft are associated with E winds at the surface, but figures S2 and S3 show dominant surface wind mode to be 70-210 or 60-180 which are also SE. Also note that last sentence in block mentions SE surface winds.

151 Does Fig S4 need to show full annual cycle?

153-154 Statement that Fig S4 "shows" that geostrophic winds dominate katabatic forcing of surface winds in peak summer kind of has to be taken on faith (no details about calculations supporting this until discussion of Figs 7 and 8 around line 390).

Can this idea be presented clearly just once in the manuscript (it does seem to be a finding authors want to stress).

167-170 It is usually risky to assume transport path based on wind direction observations at a receptor site. Do trajectories follow the contours as suggested? Strictly speaking, Fig 1b suggests that air traveling from Concordia to SP along 123 would cross at least 1 contour and flow uphill for almost the last half of the trip.

170-172 These closing sentences are problematic for at least 2 reasons. First is that heading out of SP along 110 is more "uphill" toward the central dome than just discussed, and one could imagine weak katabatic flow being influenced by the subtle topography east of SP shown in Fig 5 (and discussed later). Second is that very high NO at the same time winds and BLH are both high sort of explodes the entire conceptual model developed up to this point, so it hardly seems fair to ask the reader to take these observations as being in support of suggestion that summer time katabatic winds are over emphasized in the literature, and then dismiss them as being unusual.

184-191 Is it important to know about evolution of vortex dynamics since 1960, or are the key points the different date of breakup in the 4 study years, and the fact that the O3 hole results in higher actinic flux while the vortex is still intact?

192-199 this paragraph mainly restates comments made earlier

200-213 Of mild interest perhaps, but what is the quantitative link to NO at SP?

215-222 Nice verbal summary of Fig 8, but no link to NO at SP

223-229 Another paragraph linking large scale and BL dynamics that almost makes predictions about impact on NO at SP, but does not test them with the data at hand (note that cloudiness in the 4 years mentioned in line 224 does not directly correlate with frequency and length of high NO episodes, 2003 does have high NO and few clouds, but 1998 has most clouds and second highest NO).

230-247 Nice case study, and reassuring that it shows features observed in other

events in the 3 earlier seasons. Just seems kind of late in the manuscript to finally be getting to NO observations at SP in any detail.

290-291 Pretty sure that Berhanu et al. and Frey et al. feel these sequences are not coincidental. Beside that, they must have physical/chemical explanation, even if not all the details have been nailed down.

295-420 Please make it more clear what new insights are gained from the 2006 data set. Mostly seems to just reinforce things that were seen before.

300-302 Can't see any hash marks plotted at -20 in Fig 6.

324-338 A little unclear how the comparisons in the early part of this block lead to suggestion that late Dec emissions in 2006 were low due to photobleaching of the snow. In the day 305-340 block NO was 2 x higher in 2003 with similar wind/cloud statistics, but then in Dec the mean concentrations were similar despite 2003 being cloudier with deeper BL. Seems that maybe 2003 started with more nitrate in the snow for some reason, but in 2006 the lower amount available to make NOx kept doing so longer due to favorable BL dynamics (e.g., average decreased by factor of 2.2 in 2003 in response to seasonal change, while in 2006 the early season to late season difference was only factor of 1.4)

351 as noted earlier, seems surface winds tend to be from SE, not E, when 300 hPa winds are SE

352-353 Not very likely that much NOx is transported in FT to SP

354-356 Just mentioned these earlier studies and suggested that maybe the link was just random.

356-372 What motivates this rehash of 2003 results in a section on "Case studies and insights from new 2006 data"?

385-400 Nice development of the argument against significant katabatic forcing, but

it feels redundant since the finding was earlier declared (with little support first time around)

403-410 Here is another example of using case study to make a solid point, that was earlier just boldly declared (lines 70-74). If this is so well established that it is in introductory remarks, does it need support here? Or should the earlier section be pulled back a little.

420-446 Could/should these details be moved to supplement? Key point is to use the estimated BLH to estimate NO fluxes.

455 measurements at SP

461 associated lifetime

513-522 Think another important aspect of these dynamical findings is that none of them, alone, had strong direct correlation with NO concentrations or fluxes at SP. This point is raised several times in the rest of manuscript, why not here?

574 because of potential

647 and 649 is the "0.000 level" correct?

817 the red line in 4a is not that easy to discern from black, try lighter shade

Figure S3, red diamonds in a and b seem same maroon as in Fig 4a, very close to black. Try lighter hue

Figure S5 caption, what do you mean by "year-to-year scatter is not unreasonable"?

Figure S6, Not sure this figure is central to the NO story, but if it stays may need to say something about what is so significant about the Breakup date. In both intervals it certainly looks like hole is filling weeks earlier. Also, why is the early interval averaged 1964-1980 (dots) but break up date averaged 1961-1980?

Figure S9 caption. Figure 3 does not show any basin. Better callout would be Fig 5

(possibly Fig 1, but suspect 5 is better).

Figure S10, as noted in text, would probably be better as 4 panels. And it should be in the main paper, not supplement.

[Figure]

---

## Referee Comment (RC2) · Anonymous Referee #2 · 2 Dec 2017

This paper describes data prescribing the boundary conditions affecting the near surface air-chemistry at the South Pole; more specifically the conditions are sought that lead to occasional episodes of surprisingly high levels of NO in the lowest 50 m or so of the atmosphere.

This is a complex discussion: NO levels may depend on large scale meteorology (advected air from the oceans), small scale mixing (boundary layer stability), sunlight, and the actual chemistry sources and sinks. The paper faces a significant challenge is presenting the reader with these processes, their importance, and the supporting data (from different campaigns) in a manner that tells the story and supports the conclu-

sions. It is this challenge that I found wanting.

I think the paper is difficult to read: this may be in part because much of it is not my field, but I suggest that most readers will suffer similarly given the interdisciplinary nature of the discussion. The authors therefore need to help set the story better, and I suggest that two or more schematics would be most helpful. The source, mixing and ventilation of the boundary layer, with the chemical pathways (in snow, air and advected aloft) overlaid. This coupled to maps (as per figure 1, 5 and 8) with an overlay of wind roses, rather than x-y plots (figure 1 again). The authors should think of a clearer nomenclature for wind direction, as "157.5" and "337.5" implies a very highly modal air flow, rather than, for example "the SSE and NNW sectors" (I assume this is what the authors meant). Perhaps even include a sailors' compass for those less familiar with these terms, but emphasise that such sectors have natural angular range bin of a quarter of a right angle. These would then fit nicely with wind roses of either 8 or 16 direction bins. Finally on the topic, such schematics would stress that 'North' at the South Pole is nominal, and the meridian is taken.

The schematic of the boundary conditions would greatly assist with giving meaning (and importance) to the whole of Section 2. Each section describes some meteorological phenomenon, but not why it matters. The reader (at least this one) was left with a wealth of information dangling, without a mechanism to sift for importance for the overall Question. All of the information presented may be vital to the argument, but, I would ask the authors to check each statement here for Invasion of the Interesting Fact (which isn't actually critical).

Perhaps (again for the non-specialist reader) the conclusions could be presented as a "recipe for a perfect NO event", that is, High NO is likely to happen when (a) and (b) and (c) or (a) and (d) but not (d) etc.

---

## Author Comment (AC1) · 10 Jan 2018

Response to RC1

This manuscript focuses on the factors (mainly dynamical, with some attention to chemical processes) that contribute to large, but variable, mixing ratios of NO at South Pole (SP) during the spring to summer transition. Previously unpublished measurements of NO and NO2 during the 2006-2007 field season are presented and discussed in context of observations made in 3 earlier campaigns (1998/99, 2000/01, and 2003/04 seasons). Several previous papers by this team have established that the surprisingly high levels of NO sometimes seen at SP are sustained by snow to air flux of NO produced by photolysis of nitrate in the snow, building up when low boundary layer height (BLH) limits vertical mixing. Low BLH resulted from low wind speeds and clear skies, with the latter also enhancing the rate of nitrate photolysis. Intervals with low BLH generally coincided with easterly winds, leading to a suggestion that some of the NO during extreme events was transported down slope to SP, with extensive recycling of redeposited nitric and pernitric acids combined with non-linear NOx chemistry resulting in mixing ratios approaching and even exceeding 1 ppbv.

The addition of the 2006 results confirms earlier findings but does not produce any fundamental new understanding.

Response: We feel the value of 2006 data lies in the comparison of the behavior of NO over a greater portion of the seasonal cycle. These data are also useful to compare and contrast 2003 and 2006 which are similar in many aspects of their meteorology but quite different from 1998 and 2000. Earlier papers did not examine year-to-year variability and effects on NO. In addition, the step-wise linear regression in Table A4 shows that the dominant parameters for NO vary from year to year reflecting the variability in weather regimes affecting NO at the SP. Thus, the new finding we show is that not only many variables are important but that their relative importance varies from year-to-year.

However, this manuscript does extend prior papers by exploring the links between BL dynamics at SP and synoptic and mesoscale dynamics in the upper troposphere and even into the stratosphere over Antarctica. In my opinion the key findings are that high NO at SP are most likely to occur when the flow at 300 hPa is from SE, which is found to correlate with relatively weak surface winds from the E-S quadrant and generally clear skies, factors leading to low BLH. It is also pointed out that the south-easterly flow at the surface is due to large scale pressure gradients, and not katabatic flow as had earlier been suggested.

These findings are significant enough to merit publication, but I find the manuscript overly long, repetitive at times, and hence not as accessible as it could be.

Response: Because the paper evolved over a period of time we admit there was repetition which we have hopefully reduced. As far as the length, Doug Davis and I were originally considering two papers. However, with his untimely passing, I and the other co-authors decided it was best to complete and include as much of his material as was feasible in just one paper. We appreciate the detailed reading of the manuscript and very useful comments that this reviewer provided.

In particular,I find that a lot (too much) of the discussion relies on figures in the supplemental material, and that graphical or tabular comparisons of NO in the 4 different seasons

are never provided in the main body of the manuscript. Figure S10 does provide such an overview, but I suggest it would work better as a 4 panel plot (one panel for each season) and should be moved out of the supplement to someplace early in the main text. Current version of sections 2.1 and 2.2 go into great detail about meteorological similarities and differences across the 4 sampling seasons, but rarely relate these to observed NO concentrations at SP (which seems like it should be the main point of paper). Referring (probably often) to a revised figure S10 put into section 2 would help.

Response: We appreciate this suggestion and have now included a new figure and table near the beginning of the paper that summarize the four seasons of data and highlight the differences between the observing periods. This now allows us to motivate the discussion of the meteorology in subsequent sections.

Similarly, Fig 2 would be improved by adding some kind of NO metric (e.g., season mean, or fraction of sampling hours > than some threshold (250 pptv is used elsewhere)), and the heavy black line showing average NO as function of surface wind direction in Fig 3 now averages all 4 seasons, unnecessarily obscuring interannual variations which are important and interesting.

Response: A summary of NO metrics are now included in Table1. Fig. 3 has now been revised to show individual years in terms of average NO and wind direction distributions in a two-panel display.

 In the following list of detailed comments I mix purely editorial suggestions (typos, etc.) with comments related to streamlining the narrative by reducing redundancy and interesting side stories. Comments are linked to line numbers in the on-line pdf, hope these are conserved across different platforms.

Response: We very much appreciate the time and effort that this reviewer put into their review and suggestions below. They have been most helpful!

69-73 Would help to indicate the length of these weather cycles. Seems the timescales are a few days of clouds, then a few with clear skies and strong inversion based on case studies presented much later, but here the reader could assume you mean seasonal changes.

Response: Text added: "*These changes occur over normal synoptic time scales of a few days and over planetary wave time scales of 10-60 days (e.g. Yasunari and Kodama 1993)*."

73 I would argue that high albedo and low SZA are characteristic of most of Antarctic plateau. What is "unique" about SP is that the SZA barely changes within 24 hour day. Later on it seems the SZA is felt to be significantly lower at SP than at 75 or 80 S, but is that true averaged over a full day? Might help just to be quantitative about meaning of "low" SZA.

Response: New text has been added:
"*A detailed study of the surface energy budget and boundary layer evolution at Concordia Station (King et al., 2006) shows a convective boundary layer and positive net radiation lasting eight hours (averaged*

*over December and January).  This convective period developed when the solar elevation angle was greater than 25$^o$, in contrast to the lack of thermal convection at the SP where the maximum solar elevation angle is always less than 23.5$^o$.  Frey et al. (2015) show a specific example for 9 January at Concordia when a stable boundary layer forms with large increases in NO shortly after 1800 LT when the solar elevation angle decreases to less than 20$^o$.   Similarly King et al. (2006) show a sodar record from 28 January with a stable boundary layer until 0800 and after 1730 LT when solar elevation angle less than ~22$^o$.  For these reasons air masses arriving over long distances at the SP are likely to have encountered some convective mixing enroute and may need to be considered in future model simulations of surface chemistry.”*

78 “omnipresent” seems an odd description of the inversion, given you just said the really stable boundary layer reforms every weather cycle.

This text has been eliminated. One confusion lies in the distinction between a “temperature inversion” and a “stable boundary layer.”  Often a very shallow stable boundary layer will form within a deeper inversion layer as might be seen in a rawinsonde temperature profile.  It is only in this sublayer that vertical mixing would occur. Also, at times a very shallow convective layer can form within a deeper “inversion layer.”  This typically occurs when colder air passes over a previously ‘warmed’ surface: the equilibrium state is often one with an isothermal layer with a capping inversion (Neff, 1980).

87-88 words missing here, like “Antarctica” or the Antarctic ice sheet. Does not make sense to describe storm track as pole centric, or covering so many

Response: text removed as part of the restructuring of Section 2.

95-96 Does not seem the semi-annual oscillation or the circumpolar trough are ever mentioned again, so why introduce the terms here? Are they controlling NO at SP?

Response: new text added: *“Although there can be significant year-to-year variability in storminess, October can be the period of greatest snow accumulation (as measured in the stake field at the SP).  The average accumulation in October from 1990 to 2015 is 3 cm whereas in November it is 1.5 cm (amrc.ssec.wisc.edu/pub/southpole/climatology).   During that 25-year period, 2003 was one of the four highest accumulation years. In 2003, the snow depth increased by 6 cm in October whereas in November it decreased by 1 cm.  In our other experiment years the accumulation ranged from 1.5 cm (2006) to 4.0 cm (2000) in November.  The accumulation rate may be important to the fate of nitrate in the snow (e.g. Mulvaney et al. 1998) as well as the exposure of nitrate in the skin layer to photolysis and recycling (e.g. Frey et al. 2015)..”*

98-100 Another interesting tidbit about large scale flow in winter that seems to have nothing to do with NO at SP in Spring/Summer

Response: Text deleted.  The point was that with warm advection, clouds and higher temperatures occur (as observed in both winter and summer) with the attendant increase in boundary layer depth which is associated with low NO.

103-104 Is it fair to compare one 135 degree sector to another that is only 90 degrees? I also note that if wind direction was completely uniformly distributed the larger sector

would account for 37.5% and the W to N quadrant 25%, making the quoted 43 and 31% fractions observed only slightly enhanced. Figure 1 makes the point, so consider deleting this sentence.

Done – as pointed out by this reviewer Fig. 1 says it all with peaks in the distribution 180$^o$ apart.  Wind roses have now been added in the supplemental material to provide more detail.

109-113 These last 2 sentences repeat ideas introduced earlier (in this paragraph, also in first paragraph of section 2, and in earlier papers). It is also odd to be making predictive statements about how NO should respond, rather than point to data that confirms the points discussed. Granted, the 2006 results have not yet been show, but the other 3 campaigns are published.

Response: Text removed.

114-121 Perhaps unnecessary detail, the correlation is interesting and useful for chemists on its own.

Response: We felt this was appropriate for Supplemental Material because the figure shows how synoptic pressure patterns extending over the Plateau affect surface flows which are important to boundary layer chemistry.

121 can, however, vary—-! Varies

Response:  corrected

124 mention 1999 and 2002 and say they coincide with a single year with sudden strat warming. Which one had the warming?

Response: 2002 is now indicated as the year with sudden stratospheric warming (in September).

125 Any decadal periodicity is not at all obvious in Fig 2. Even if it is there, is it important to understanding any differences between the 4 years (spread over 9 years) with observations at SP.

Response: The data are now shown with a five-year smoothing.  As now noted in the text, 2002 was eliminated based on the unusual breakup of the vortex that spring and its potential impact on the troposphere (e.g. Charlton et al 2005).   The semi-decadal variability is important for putting any particular year in a longer term context.  Reference is now made to Neff (1999) for a past study of decadal variability (in that paper the focus was on decadal variability in cloud fraction).

[Figure]

**Figure 11.** Frequency of occurrence (FOO) (in percent) of upper level winds from south-southeast (SSE) (summed over 90°, solid line), west-northwest (WNW) (summed over 90°, short-dashed line), together with the minima in the 10.7-cm solar radio flux (arrows) estimated from data obtained at ftp://ftp.ngdc.noaa.gov/STP/ SOLARDATA/SOLAR_RADIO /FLUX).

[Figure]

**Plate 7.** Frequency of occurrence (percent in each 22.5° class interval, 0.5% contour intervals from 5% to 10%) of upper level winds for the spring period shown in Plate 5. Upper level (~300 hPa) winds were sorted by year and by 22.5° class intervals and then smoothed with a 5-year tapered interseasonal filter and a three-point tapered class interval filter. Class intervals were centered on the directions beginning at 0°. The band of directions from 180° to 270° corresponds to the direction of maximum warm advection (Figure 7). Comparing these results with those in Plate 5 reveals that periods of maximum (minimum) cloudiness occur during maximum (minimum) frequency of occurrence of winds from this direction. Dashed lines and labels indicate the times of transition through 225°. Missing rawinsonde data from 1957 through 1960 make the first time of transition uncertain by a few years.

129-130 Is it really a new finding that NO at SP is not tightly controlled by any single parameter? Seems this has been noted in earlier papers.

Response: Earlier papers did not examine year-to-year variability and effects on NO.  In addition, the step-wise linear regression in Table A4 shows that the dominant parameters for NO vary from year to year reflecting the variability in weather regimes affecting NO at the SP. Thus, the new finding we show is that not only many variables are important but that their relative importance varies from year-to-year. This has been added to the Introduction:  "*However, earlier work did not examine year-to-year meteorological variability and effects on NO.  In addition, the step-wise linear regression that we show later demonstrates that the dominant parameters for NO vary from year to year reflecting the variability in weather regimes affecting NO at the SP. Thus, the new finding we will show is that not only many variables are important but that their relative importance varies from year-to-year.  To address some of these processes, we analyze previous data sets complemented by a new data set collected in 2006-07 that extends over much of the seasonal cycle.  Past work emphasized the relationship between high NO and shallow boundary layers (Davis et al. 2008; Neff et al. 2008).   Here, the primary focus is on providing a broader perspective of the origin and evolution of high NO episodes, their relationship to stable boundary layer meteorology, the unique large-scale meteorology of the high Antarctic plateau, the influence of the changes in the stratosphere in recent decades, and the effect of variability within seasons as well as from year-to-year.*"

134-170 A little confusing here, with first sentence saying SE winds aloft are associated with E winds at the surface, but figures S2 and S3 show dominant surface wind mode to be 70-210 or 60-180 which are also SE. Also note that last sentence in block mentions SE surface winds.

Response:  This has been addressed with an addition to Fig. 3 that shows the relationship of 300 hPa and surface winds. New text: "*Figure 3a shows the distribution of surface wind directions for each experiment year as well as the average NO for each year as a function of surface wind direction in Fig 3b. Although the wind distributions are similar for both 2003 and 2006, the average NO is much less in 2006. In Fig. 3b, highest NO occurs with surface winds with an easterly component (between $60^o$ and $180^o$.)  In Fig. 3c, for the entire period 1961-2010, we show the wind distribution at 300 hPa for surface winds between $60^o$ and $180^o$ and the distribution of surface winds in particular when the wind at 300 hPa is between $120^o$ and $150^o$.  The distribution is fairly broad (with a median of $87^o$ and the peak centered on $105^o$).  The breadth of this distribution may be related under sampling with once- or twice-a-day rawinsondes and surface observations.*"   This latter comment is relevant to Fig. 7 where we show a case where the surface wind direction shift occurs after the rawinsonde launch but before the synoptic time reported for the rawinsonde.  The launch on Day 337 in Fig. 7 was at 0953Z (prior to the wind shift) but reported for 1200Z by which time the surface wind had shifted to $135^o$ from $349^o$ at the time of the launch.

It should be noted that in the paper we have used the corrected rawinsonde wind data rather than that reported in the GTS where wind directions from $75^o$ to $315^o$ were largely missing (Corrected data are available from ftp://amrc.ssec.wisc.edu/pub/southpole/radiosonde/).  Errors in the processing software in the field produced erroneous winds (from 14 February 2005 until 16 June 2007) that were later corrected (https://amrc.ssec.wisc.edu/presentations/neff_etal_rawin.pdf*)*.

151 Does Fig S4 need to show full annual cycle?

Response: Insofar as we are only interested in summer regimes, we do not feel it is necessary beyond showing the timing of the transition to and from summer regimes.

153-154 Statement that Fig S4 "shows" that geostrophic winds dominate katabatic forcing of surface winds in peak summer kind of has to be taken on faith (no details about calculations supporting this until discussion of Figs 7 and 8 around line 390). Can this idea be presented clearly just once in the manuscript (it does seem to be a finding authors want to stress).

Response: New text has been added.  Reference to Neff 1999, Sec. 4.2 is now made where this is discussed as well as a later reference to Parrish and Cassano (2003) where a model study confirmed the result. It is also noted that Ball's (1960) nomogram comparing horizontal pressure gradients to those due to katabatic ones was adapted by Neff (1980) in terms of geostrophic wind speeds which is the basis of the results in Fig. S4. The importance of this result is that accumulation pathways for NO in the surface flows cannot be tied to simple concepts of gravity driven flows over a sloped surface.

167-170 It is usually risky to assume transport path based on wind direction observations at a receptor site. Do trajectories follow the contours as suggested? Strictly speaking, Fig 1b suggests that air traveling from Concordia to SP along 123 would cross at least 1 contour and flow uphill for almost the last half of the trip.

Response:  This text has been deleted and the discussion is now presented in the context of Figs. 5, 8, and S13 (which shows a back trajectory on 30 November 2003 from ERA-I) which suggest much more complex mesoscale circulations over the high plateau.

170-172 These closing sentences are problematic for at least 2 reasons. First is that heading out of SP along 110 is more "uphill" toward the central dome than just discussed, and one could imagine weak katabatic flow being influenced by the subtle topography east of SP shown in Fig 5 (and discussed later). Second is that very high NO at the same time winds and BLH are both high sort of explodes the entire conceptual model developed up to this point, so it hardly seems fair to ask the reader to take these observations as being in support of suggestion that summer time katabatic winds are over emphasized in the literature, and then dismiss them as being unusual.

Response:  This case is an outlier in some respects.  Initially the boundary layer was quite shallow and then as winds increased at times it became deeper with high NO.  Unfortunately, we have no observations upstream to diagnose this case or to identify an accumulation area upstream. In addition, the meteorological tower data was missing from Days 333-334.  The independent AWS station during this period did show consistent winds from $120^o$ which is not a katabatic wind direction.   We do point out later that an aircraft flight on 4 December (Davis et al. 2008) after the end of the event did show large NO between 100 km and 400 km from the SP along $110^o$  even as concentrations were dropping at the SP. This suggests the possibility of inhomogeneous accumulation areas on the plateau.

Another factor that we have not discussed in this paper is the potential role of boundary layer convergence due to cross-isobar Ekman layer flow in the boundary layer of  mesoscale low pressure systems.  Neff (1980) calculated the e-folding time for such a stratified spin-down  process at 1.1 days.

This would be worth exploring with a high-resolution model and an enhanced observation network on the high plateau.

With respect to our assertion of flow relative to local topography we show below a more detailed look at the topography after a bit of smoothing.  As can be seen here 110$^{o}$ is more aligned along the large scale terrain contours.  In addition, with the Coriolis deflection to the left, any downslope flow from the high dome area would be away from the SP not towards.  In our later discussion we estimate the relatively weak katabatic influence of local terrain near the SP.

[Figure]

184-191 Is it important to know about evolution of vortex dynamics since 1960, or are the key points the different date of breakup in the 4 study years, and the fact that the O3 hole results in higher actinic flux while the vortex is still intact?

Response: Text added in the Introduction: "*Past work (Neff, 1999) found evidence for the effect of stratospheric ozone depletion on the tropospheric circulation over the interior of Antarctica.  A key question then was the potentially combined effects of changes in the radiative environment (via UV photolysis) and concomitant changes in the near-surface circulation affecting NO.*"

192-199 this paragraph mainly restates comments made earlier

Response: Text has been revised.

200-213 Of mild interest perhaps, but what is the quantitative link to NO at SP?

Response:  This figure shows that the characteristics of winds in the late spring have not changed but that there has been a significant change in the early summer period that coincides with increased cloudiness (and hence deeper more well-mixed boundary layers associated with lower NO) that has been confirmed through the 1990s (Neff, 1999) and for more recent periods in Fig. S8.

215-222 Nice verbal summary of Fig 8, but no link to NO at SP

Response: Link is now made: "*The minimum in cloudiness during Julian Days 325-350, when shallow boundary development is likely due to increased clear-sky radiative cooling and coincides with increased actinic flux due to the persistent ozone hole of recent decades.  This is also the period in 2003 and 2006 when the highest and most persistent NO occurred (in 1998, the field program started late and in 2000, the ozone hole filled early).  Conversely, the summer period has, on the average, become cloudier, albeit subject to inter-annual and decadal variability, and hence less conducive to higher NO.*"

223-229 Another paragraph linking large scale and BL dynamics that almost makes predictions about impact on NO at SP, but does not test them with the data at hand (note that cloudiness in the 4 years mentioned in line 224 does not directly correlate with frequency and length of high NO episodes, 2003 does have high NO and few clouds, but 1998 has most clouds and second highest NO).

Response:  We agree this paragraph was too general.  We have eliminated much of the existing text and have added text focusing on the lack of extreme values and the importance of intermittency in conditions of high NO: "*In 1998, the late spring period was particularly cloudy compared to average conditions.  However, after a late start to observations (see Fig. 1), higher NO occurred intermittently later in December during days with clear skies, light winds and higher actinic flux (due to the delay in the breakup of the ozone hole that year.  The intermittency in 1998 is also reflected in the very limited values of NO in excess of 500 pptv (only 4% compared to 25% in 2003, Table 1).*"

230-247 Nice case study, and reassuring that it shows features observed in other events in the 3 earlier seasons. Just seems kind of late in the manuscript to finally be getting to NO observations at SP in any detail.

Response: This case now follows as a specific example pulling together the roles of intermittency relative to the late-spring climatological minimum in cloudiness, mesoscale eddies, and changes in total column ozone.

290-291 Pretty sure that Berhanu et al. and Frey et al. feel these sequences are not coincidental. Beside that, they must have physical/chemical explanation, even if not all the details have been nailed down.

Response: Unfortunately, these two papers as part of the OPALE program had different foci.  In Berhanu et al, the interest in wind-blown snow occurred in the context of changes in isotopic composition in their samples whereas Frey et al. focused on explaining $NO_X$ levels and effects of mixing processes as well as modeling the flux of $NO_X$ from the snow.
Because of the typical decrease in wind speed from November to December, we were curious whether the mechanism described my Mulvaney was also at work on the high plateau.  We feel this is something worth exploring in future work.
We have reworded this last section as follows: "*Recent research at Concordia Station on the high Antarctic Plateau reported a similar increase in surface nitrate in early December (Berhanu and Co-authors 2015).  Wind speed had been high in late November (Frey et al. 2015, Fig. 1) followed a week later by high skin nitrate and high fluxes of $NO_X$ from the surface consistent with the observations of Mulvaney et al. (1998) at Halley Station.*"

295-420 Please make it more clear what new insights are gained from the 2006 data set. Mostly seems to just reinforce things that were seen before.

Response:  We added an overview of the four episodes in Fig. 1 to set the stage for the subsequent discussion:

"*The first three field seasons have been described in some detail (Davis et al 2001, 2004, 2008) whereas a fourth more extensive observation of the seasonal cycle in NO, obtained in 2006-2007, has not yet been published.  The four seasons of NO data are shown in Fig. 1 and their statistics in Table 1.  In 1998, the ozone hole broke up very late as seen in Fig. 1a but the observational program was also delayed until early December. In 2000 (Fig. 1b), total column ozone increased well before the beginning of the field season and NO remained relatively low. In both 2003 (Fig. 1c) and 2006, the ozone hole broke up about the same time but higher column ozone was more persistent in 2006 until late December.  In Fig. 1d, we also show the solar elevation angle (x30) increasing from $14.5^o$ to $23.5^o$ as well as the nitrate photolysis rate $J(NO3^-)$ which initially follows the seasonal increase in elevation angle but responds after JD330 to shorter term variations in total column ozone and meteorology.  In both 2003 and 2006, prolonged periods of high NO concentrations occur prior to Julian Day (JD) 340. In particular, in 2003, the period of JD 326-336 has a median concentration of 482 pptv whereas the sustained period in 2006, JD322-331 has a median value of 389 pptv.  After JD340 the median values are 247, 91, 152, and 146 pptv for each of the four seasons respectively.  Higher NO values in 1998 after JD340 presumably reflect the delay in the breakup of the ozone hole.  In Table 1, we summarize the occurrence of NO concentrations in three ranges for each of the field seasons.  2003 stands out as having the highest number of NO concentrations exceeding 500 pptv, 90% of which occurred before JD 335 while 2000 had the fewest.  Also 1998 saw the fewest hours of NO exceeding 500 pptv.  From the perspective of Fig. 1, the first three years produced diverse results.  The additional data in 2006, while more extensive, confirmed the general seasonal pattern seen in 2003.*"

The first two field seasons as we have noted were unusual with 1) a delay in starting observations in 1998 and then in 2000 a very early breakup of the ozone hole and subsequent low NO levels and few winds with an easterly component.  The data in 2006 provided an opportunity to describe the full spring-summer cycle in NO.

300-302 Can't see any hash marks plotted at -20 in Fig 6.

Response: Revised the figure to only show clear sky periods as indicated by high direct radiation. Plotted every fourth data point for clarity.

324-338 A little unclear how the comparisons in the early part of this block lead to suggestion that late Dec emissions in 2006 were low due to photobleaching of the snow. In the day 305-340 block NO was 2 x higher in 2003 with similar wind/cloud statistics, but then in Dec the mean concentrations were similar despite 2003 being cloudier with deeper BL. Seems that maybe 2003 started with more nitrate in the snow for some reason, but in 2006 the lower amount available to make NOx kept doing so longer due to favorable BL dynamics (e.g., average decreased by factor of 2.2 in 2003 in response to seasonal change, while in 2006 the early season to late season difference was only factor of 1.4)

Response:  This section has been rewritten to avoid the confusion:

*"Comparing these two years reveals the complexity introduced by meteorology: In 2003 NO was fairly low after JD340 during a period of higher winds and cloudy conditions but then followed by a single high NO episode near the end of the month whereas 2006 saw moderate NO concentrations intermittently throughout the period (see Fig. X).   Thus the much lower average NO after JD340 in both years  appear consistent with the conclusion from (Frey et al. 2015b) for the last half of December and early January at Concordia Station.  Frey et al. argues, based on the laboratory results of (Meusinger et al., 2014) that the snow nitrate is composed of both photo-stable and photo-labile fraction and that the photo-labile fraction decreased later in December at Concordia Station. Our results would be consistent with that interpretation and show the value of multi-year and intraseasonal comparisons of NO."*

351 as noted earlier, seems surface winds tend to be from SE, not E, when 300 hPa
winds are SE

Response: This has been addressed by modifications to Fig. 3 and associated text.

352-353 Not very likely that much NOx is transported in FT to SP

Response: We have softened this assertion with the following text: "Past work has indicated that the SP is not isolated from the influence from more northerly latitudes.  For example,  high methanesulfonate was observed from 28 November to 2 December -- an indication of marine sources (Arimoto et al. 2008). This period had the highest NO values recorded in the entire series of field programs (Davis et al. 2008) similar to those found at Concordia in a similar time frame in 2011 (Frey et al. 2015).  Furthermore, as suggested in (Arimoto et al. 2008)  air arriving at the South Pole may have had its origins in continental regions (due to much higher concentrations of the elements Pb, Sb, and Zn than in previous field programs) as well as in the ocean off Wilkes Land on the far side of the continent."

354-356 Just mentioned these earlier studies and suggested that maybe the link was
just random.

Response: This paragraph was revised as above..

356-372 What motivates this rehash of 2003 results in a section on "Case studies and
insights from new 2006 data"?

Response: Changed the title: Case studies and insights from new 2006 data in the context of past experiments. We have also pointed out the differences in seasonal snow accumulation rates between 2003 and 2006 as well as highlighting the similarities and differences in the meteorology between the two years.

385-400 Nice development of the argument against significant katabatic forcing, but it feels redundant since the finding was earlier declared (with little support first time
around)

Response:  Earlier discussion was expanded using Ball (1960) and Neff (1980).  The rationale here was to look on potential local effects due to the terrain less than a 100 km from SP not the large-scale terrain.

403-410 Here is another example of using case study to make a solid point, that was earlier just boldly declared (lines 70-74). If this is so well established that it is in introductory remarks, does it need support here? Or should the earlier section be pulled back a little.

Response: Earlier discussion was expanded. This discussion was pulled back a bit but we still think the analogy between the rapid onset of a stable boundary layer at Concordia due to low sun elevation in the diurnal cycle and the rapid onset at SP due to increased radiative losses following clear skies is worthwhile.

420-446 Could/should these details be moved to supplement? Key point is to use the estimated BLH to estimate NO fluxes.

Response: The last paragraph has been removed as it duplicates earlier text. The discussion of Fig. 10 is useful because it provides a comparison of BLD versus NO for all four years and implies variability in source/flux terms that comes next.

455 measurements at SP

Corrected

461 associated lifetime
Corrected

513-522 Think another important aspect of these dynamical findings is that none of them, alone, had strong direct correlation with NO concentrations or fluxes at SP. This point is raised several times in the rest of manuscript, why not here?

Response: We have added the following text: "This case also reveals the difficulty in correlating peaks in NO directly with upper-level winds insofar as the peak in NO occurs in the gap between rawinsonde observations. Perhaps more important are these transitions in upper level winds that signify a direct connection to large scale dynamical changes in the atmosphere over the plateau. "
We also added in the discussion of Fig. 7 the following text: "Figure 7 also suggests limitations in correlating surface NO with wind shifts at 300 hPa: In this case the peak in NO occurs before the shift in winds aloft; by the next rawinsonde 12 hours later NO has dropped substantially."

574 because of potential collinearity

Added phrase: "such as between wind direction and temperature or wind speed and temperature"
And added: "This approach eliminates redundant variables."

647 and 649 is the "0.000 level" correct?

Response: Yes. Changed text to "highly significant at the 0.000 level (using SPSS software)"

817 the red line in 4a is not that easy to discern from black, try lighter shade
Figure S3, red diamonds in a and b seem same maroon as in Fig 4a, very close to

black. Try lighter hue

Response: Done (this seems really sensitive to particular monitors/printers).

Figure S5 caption, what do you mean by "year-to-year scatter is not unreasonable"?

Response: reworded to be more explicit: : "The differences in timing determined from Harnik et al. (based on time of circumpolar stratospheric winds reversing) and the time of formation of the thermal tropopause (from the SP rawinsonde) is quite reasonable."

Figure S6, Not sure this figure is central to the NO story, but if it stays may need to say something about what is so significant about the Breakup date. In both intervals it certainly looks like hole is filling weeks earlier. Also, why is the early interval averaged 1964-1980 (dots) but break up date averaged 1961-1980?

Response:  The Dobson data only started in 1964.  New text added:
"The reduction in ozone now coincides with the period most favorable to SE winds shown in Fig. 1 (with clear skies and shallow boundary layers important to high NO concentrations).  In addition, changes shown in Fig. S6 suggest long term increases in actinic flux with a 30% reduction in total column ozone by the average time of vortex breakup in recent decades. In addition, the largest changes occur between JD320 and JD330 corresponding to the periods of highest NO in 2003 and 2006 as shown in Fig.1."

Figure S9 caption. Figure 3 does not show any basin. Better callout would be Fig 5 (possibly Fig 1, but suspect 5 is better).

Response: This should be Fig. 5 – failed to update figure numbering after earlier changes in the paper.

Figure S10, as noted in text, would probably be better as 4 panels. And it should be in the main paper, not supplement.

Response: done

---

## Author Comment (AC2) · 10 Jan 2018

This paper describes data prescribing the boundary conditions affecting the near sur- face air-chemistry at the South Pole; more specifically the conditions are sought that lead to occasional episodes of surprisingly high levels of NO in the lowest 50 m or so of the atmosphere.

This is a complex discussion: NO levels may depend on large scale meteorology (ad- vected air from the oceans), small scale mixing (boundary layer stability), sunlight, and the actual chemistry sources and sinks. The paper faces a significant challenge is presenting the reader with these processes, their importance, and the supporting data (from different campaigns) in a manner that tells the story and supports the conclu-
sions. It is this challenge that I found wanting.

Response: We have made a number of revisions in response to RC1 including a four-season overview at the beginning of Section 2 and a figure that summarizes the NO observations for all four years and shows the variability in actinic flux, solar elevation angle and timing the breakup of the ozone hole for 2006-2007. This then sets the stage for the subsequent discussion in Section 2.

I think the paper is difficult to read: this may be in part because much of it is not my field, but I suggest that most readers will suffer similarly given the interdisciplinary nature of the discussion. The authors therefore need to help set the story better, and I suggest that two or more schematics would be most helpful. The source, mixing and ventilation of the boundary layer, with the chemical pathways (in snow, air and advected aloft) overlaid. This coupled to maps (as per figure 1, 5 and 8) with an overlay of wind roses, rather than x-y plots (figure 1 again).

Response: We have now included reference to Davis et al. (2008, esp. his Fig. 2) which outlines many of the processes at work on the high plateau. New text added in the introduction includes:

"*Past work (Neff, 1999) found evidence for the effect of stratospheric ozone depletion on the tropospheric circulation in the Austral spring over the interior of Antarctica. A key question then was the potentially combined effects of changes in the radiative environment (via UV photolysis) and concomitant changes in the near-surface meteorology affecting NO. The complexity of all the potential processes affecting NO are well captured graphically in Davis et al. (2008, Figure 2) where they identify atmosphere-surface exchange processes, plateau drainage, continental outflow, lower latitude transport, and boundary layer-free troposphere exchange as the key meteorological processes.*"

As noted above we have set the stage for the discussion in Section 2 with a new figure summarizing the year-to-year differences in the four field programs. We find the x-y plots to be a good short summary of the intra-seasonal changes in the larger scale circulation but have added wind roses in the supplemental material that highlight the changes in more detail than in our new Fig. 2. This new supplemental figure also summarizes the changes in cloud fraction versus 300-hPa wind direction through the three subseasons.

The authors should think of a clearer nomenclature for wind direction, as "157.5" and "337.5" implies a very highly modal air flow, rather than, for example "the SSE and NNW sectors" (I assume this is what the authors meant). Perhaps even include a sailors' compass for those less familiar with these terms, but emphasise that such sectors have natural angular range bin of a quarter of a right angle. These would then fit nicely with wind roses of either 8 or 16 direction bins. Finally on the topic, such schematics would stress that 'North' at the South Pole is nominal, and the meridian is taken.

Response: As noted above, we have added wind roses in the supplemental material and changed angular descriptions to the SSE etc nomenclature suggested by the reviewer in Fig. 1. We agree with the reviewer that wind roses reveal the angular ranges more explicitly. In the supplemental figure we have also added wind direction/cloud fraction roses. These show that low cloud fraction in the period JD310-340 can be associated with a much narrower range of directions of 300 hPa winds (SE to ESE) than might be concluded from just wind roses. These also confirm our assertion that higher cloud events are associated with winds aloft from the direction of west Antarctica while there is a major shift in cloud/wind direction in early December

The schematic of the boundary conditions would greatly assist with giving meaning (and importance) to the whole of Section 2. Each section describes some meteorolog- ical phenomenon, but not why it matters. The reader (at least this one) was left with a wealth of information dangling, without a mechanism to sift for importance for the overall Question. All of the information presented may be vital to the argument, but, I would ask the authors to check each statement here for Invasion of the Interesting Fact (which isn't actually critical).

Response: Section 2 has been significantly revised in response to RC1 which also addresses this comment.

Perhaps (again for the non-specialist reader) the conclusions could be presented as a "recipe for a perfect NO event", that is, High NO is likely to happen when (a) and (b) and (c) or (a) and (d) but not (d) etc.

Response: Unfortunately we have found there is no simple recipe for a perfect NO event.  In fact our stepwise linear regression in Appendix A shows different variables dominate each of the four seasons that we studied.  We have revised the first few paragraphs of our conclusions as follows to make this point clearer:

"*Earlier work (Davis et al. 2008; Neff et al. 2008), primarily based on 2003 data which included direct sodar measurements of boundary layer depth,  presented a straightforward conceptual model that linked high NO to the presence of shallow boundary layers, light winds, and stronger surface inversions. In our examination of four seasons of observations, explanation of the initiation and evolution of high NO episodes as well as intra-seasonal to interannual variability proved more challenging.   Using four spring-to-summer seasons of observations, we have described the influence of the synoptic- to-mesoscale weather patterns and their seasonal cycle on stable boundary layer characteristics at the South Pole in the spring-summer period that set the stage for high NO episodes.  These included*

*1) The relative unimportance of katabatic forcing compared to the accelerations due to synoptic and mesoscale scale pressure gradients from November through January. In fact, visualizations of near-surface airflow using ERA-I (Dee et al. 2011) revealed complex mesoscale circulations that belied any simple explanation of accumulation pathways for NO.*

*2) The effect of clearing skies locally that led to rapid radiative losses and the formation of very shallow inversion/boundary layers and high NO.   Given observations only at the SP, the geographical extent of such radiation-driven boundary layers is unknown but worthy of further field observations.  Unfortunately aircraft measurements of NO in 2003 (Davis et al. 2008) were not permitted between $-40^{o}W$ and $120^{o}$ E (the clean air sector), an area which encompasses a large almost horizontal plane extending four hundred km east of the SP. However, the one flight along $120^{o}E$ at the edge of this plateau showed NO in excess of 400 pptv between 100 and 400 km from the SP at the same time NO concentrations were dropping at the SP.*

*3) The three-phase transition in spring for 300-hPa winds over the SP. These three phases corresponded to a) a late winter regime of transport of moisture over west Antarctica to the interior when the circumpolar trough is at its maximum and opens the possibility for the transport of NO precursors from northerly latitudes, b) an early spring semi-bimodal regime with 300 hPa winds alternating between northwest and southeast quadrants during November and early December as part of the seasonal cycle, followed by c) an early summer regime favoring 300-hPa winds from the Weddell Sea and warmer cloudy conditions. During the second phase, 300-hPa winds from the southeast favored clear skies, light surface winds and shallow inversions conducive to high NO concentrations at the same time the total column ozone was still low allowing higher actinic fluxes.  300-hPa winds from the northwest favored warm-air advection and cloudy conditions resulting in deep boundary layers and low NO concentrations.*"

---

## Author Comment (AC3) · 10 Jan 2018

[revised manuscript text omitted]
.  During this period it has been noted that atmospheric nitrate typically increases in mid-October whereas surface nitrate increases later in mid-November  (Erbland et al. 2013).  Erbland et al (2015) also suggest, as an estimate, that half of the annual average nitrate comes from stratospheric sources and half from long-range transport   Meanwhile, it has been suggested that increased surface nitrate at Concordia Station in 2006 has statospheric origins (Traversi et al. 2014).  In fact, back trajectories for this period show them passing just to the east of SP in early November 2006 with higher nitrate observed on 13 November at Concordia Station whereas during our last observational field program we observed higher NO on 18 November at SP.

[revised manuscript text omitted]

---

## Author Comment (AC4) · 10 Jan 2018

[Figure]

Supplemental Figures

[Figure]

Fig S1. a) Distribution of wind speed and direction at 300 hPa by subseason in 22.5° bins (Note Fig. 2 uses 45° bins), b) Distribution for mostly overcast conditions ($\geq$8/10ths cloud fraction) by 300 hPa wind direction within each subseason. c) Distribution mostly mostly clear conditions ($\leq$2/10ths cloud fraction) by 300-hPa wind direction by subseason. Daily observed average cloud fraction was used: For extreme high and low cloud fraction this provides reasonable assurance of the observed cloud fraction persisted for both rawinsonde launches. Wind data were restricted to 300-hPa wind speeds > 5 ms$^{-1}$. Period of data was 1991-2013. The period October-December generally had twice-a-day rawinsondes. Percentages are based on the total number of observations in each ~30-day interval.

[revised manuscript text omitted]